# Circulating miRNAs in Small Extracellular Vesicles Secreted by a Human Melanoma Xenograft in Mouse Brains

**DOI:** 10.3390/cancers12061635

**Published:** 2020-06-19

**Authors:** Loredana Guglielmi, Marta Nardella, Carla Musa, Ingrid Cifola, Manuela Porru, Beatrice Cardinali, Ilaria Iannetti, Chiara Di Pietro, Giulia Bolasco, Valentina Palmieri, Laura Vilardo, Nicolò Panini, Fabrizio Bonaventura, Massimiliano Papi, Ferdinando Scavizzi, Marcello Raspa, Carlo Leonetti, Germana Falcone, Armando Felsani, Igea D’Agnano

**Affiliations:** 1Institute for Biomedical Technologies (ITB), CNR, 20090 Segrate, Italy; loredana.gug@gmail.com (L.G.); ingrid.cifola@itb.cnr.it (I.C.); laura.vilardo@itb.cnr.it (L.V.); 2Department of Neurosciences, Unit of Neuromuscular and Neurodegenerative Disorders, Bambino Gesù Children’s Hospital, IRCCS, 00146 Rome, Italy; marta.nardella@opbg.net; 3Institute of Biochemistry and Cell Biology (IBBC), CNR, 00015 Monterotond, Italy; carla.musa85@gmail.com (C.M.); beatrice.cardinali@cnr.it (B.C.); ilakiannetti@gmail.com (I.I.); chiara.dipietro@cnr.it (C.D.P.); fabrizio.bonaventura@emma.cnr.it (F.B.); ferdinando.scavizzi@cnr.it (F.S.); marcello.raspa@cnr.it (M.R.); germana.falcone@cnr.it (G.F.); 4UOSD SAFU–IRCCS-Regina Elena Cancer Institute, 00168 Rome, Italy; manuela.porru@ifo.gov.it (M.P.); carlo.leonetti@ifo.gov.it (C.L.); 5EMBL Mouse Biology Unit, 00015 Monterotondo, Italy; giuliabolasco@hotmail.com; 6Fondazione Policlinico Universitario A. Gemelli IRCSS, 00168 Rome, Italy; vplabcemi@gmail.com (V.P.); massimiliano.papi@unicatt.it (M.P.); 7Istituto di Fisica, Università Cattolica del Sacro Cuore, 00168 Rome, Italy; 8Laboratory of Cancer Pharmacology, Department of Oncology, Istituto di Ricerche Farmacologiche Mario Negri IRCCS, 20156 Milan, Italy; nicolo.panini@marionegri.it; 9Genomnia s.r.l., 20091 Bresso, Italy; felsani.armando@genomnia.it

**Keywords:** melanoma, circulating miRNAs, small extracellular vesicles

## Abstract

The identification of liquid biomarkers remains a major challenge to improve the diagnosis of melanoma patients with brain metastases. Circulating miRNAs packaged into tumor-secreted small extracellular vesicles (sEVs) contribute to tumor progression. To investigate the release of tumor-secreted miRNAs by brain metastasis, we developed a xenograft model where human metastatic melanoma cells were injected intracranially in nude mice. The comprehensive profiles of both free miRNAs and those packaged in sEVs secreted by the melanoma cells in the plasma demonstrated that most (80%) of the sEV-associated miRNAs were also present in serum EVs from a cohort of metastatic melanomas, included in a publicly available dataset. Remarkably, among them, we found three miRNAs (miR-224-5p, miR-130a-3p and miR-21-5p) in sEVs showing a trend of upregulation during melanoma progression. Our model is proven to be valuable for identifying miRNAs in EVs that are unequivocally secreted by melanoma cells in the brain and could be associated to disease progression.

## 1. Introduction

The incidence of malignant melanoma is rapidly increasing worldwide. Primary cutaneous melanoma can be managed by surgery, but metastatic melanoma often spreads to the brain (40–50% of patients with advanced melanoma), where it is notoriously difficult to treat [1,2,3]. Patients with melanoma brain metastases have a dismal prognosis, with a median overall survival of only 6 months [4,5,6,7].

In this scenario, the identification of new, easily manageable biomarkers, which reliably reflect metastatic disease, remains a major challenge that could help in integrating prognoses and improving diagnoses.

MicroRNAs (miRNAs) are short (19–24 nt), single-stranded endogenously non-coding small RNAs that act as post-transcriptional regulators of gene expression via binding to the 3′-untranslated region (UTR) of target miRNAs [8]. They play important roles in cell differentiation, proliferation and self-renewal by modulating gene expression in many homeostatic processes and pathological conditions [9,10,11]. Because of the wide range of cellular processes it is able to influence, miRNA dysregulation is commonly linked to the alteration of several key cellular pathways responsible for cancer malignant progression, including cellular invasion, migration, proliferation, angiogenesis, replicative immortality, immune evasion, and the avoidance of senescence and apoptosis [12]. The alteration of single miRNA expression is often associated with melanoma progression [13,14]; nonetheless, the extent of miRNA dysregulation in melanoma and the emergence of many miRNA families suggest that single miRNAs may exert only minimal effects. Therefore, comprehensive and integrative approaches are needed to fully understand the role of miRNAs in melanoma tumorigenesis and their therapeutic potential.

Recent studies have revealed that miRNAs are secreted, freely or as extracellular vesicle (EV) cargo, into various body fluids and function as signaling molecules to influence the recipient cell phenotypes. They can mediate the interaction between cancer cells and their microenvironment in tumor- and metastasis-promoting processes [15,16,17,18]. Tumor cells have been shown to produce large amounts of EVs, which are released in the biological fluids and contain many bioactive molecules such as lipids, proteins, and nucleic acids, including miRNAs [19,20,21,22]. EVs properly serve as reservoirs for tumor biomarkers, potentially allowing for the early diagnosis of cancer. In particular, miRNAs contained in these vesicles are stable, easily accessible, and usually reflect the pathophysiological state of the primary affected tissue. The isolation of miRNAs from EVs to identify molecular signatures for early tumor diagnosis shows great potential in the liquid biopsy landscape, which could possibly replace the more costly and invasive tissue biopsies in the near future.

In this paper, we investigated the expression profile of circulating miRNAs in an in vivo orthotopic xenograft model, injecting a human metastatic melanoma cell line in the brain of athymic (nude) mice, with the aim of simulating melanoma metastasis in the brain. We identified the 25 most abundant miRNAs contained in the small EVs (sEVs) released by tumor cells in the mouse bloodstream. By comparing the sEV-associated miRNAs found in our xenograft model with data of EV-associated miRNAs identified in human clinical cases of metastatic melanoma deposited in the Gene Expression Omnibus (GEO) repository, we found that 20 out of the 25 most abundant miRNAs detected in the mouse plasma sEVs were also expressed in EVs from Stage IV metastatic melanoma cases. Moreover, three of them showed a trend of upregulation during disease progression from healthy controls to primary and metastatic melanoma cases.

## 2. Results

To identify the circulating miRNAs secreted unequivocally by tumor cells in the blood, we intracranially injected a human metastatic melanoma cell line simulating melanoma metastasis in the brain into athymic (nude) mice. We used the M14 metastatic melanoma cell line known to show some characteristics that are prerequisites for brain metastasis formation. M14 cells were transfected with a pcDNA3-luc plasmid (M14-LUC stable cell line) to monitor tumor growth in the brain. We followed the tumor growth for up to 23 days after tumor cell injection by bioluminescence analysis (Figure 1A,B).

The histochemical examination of the tumors excised from the brain confirmed the tumor growth in the brain cortex and showed evidence of infiltration raising from the site of injection, together with the presence of tumor cellular elements throughout (Figure 1C). Overall, the tumor features observed are representative of the histological phenotype of human melanoma metastases in the brain.

We analyzed circulating miRNAs directly extracted from the total plasma obtained from melanoma-bearing mice at 7, 14 and 23 days upon tumor cell injection in the brain, as well as the miRNAs extracted from plasma-purified small EVs (sEVs) at day 23.

Small EVs released by melanoma cells were first identified and characterized in vitro after purification from M14-LUC cell culture medium. Transmission electron microscopy (TEM) revealed that all the analyzed vesicles were significantly smaller than 200 nm and showed a cup-shaped morphology characteristic of sEVs (Figure 2A).

Dynamic light scattering (DLS) was used to evaluate sEV size distribution, zeta-potential, and to quantify their concentrations. A representative radius distribution and zeta-potential distribution of the cell culture-released sEVs are reported in Figure 2B (left and right panel, respectively). Our results show that sEV preparations contain vesicles with an average radius of 52 nm and an average zeta potential of −19 mV, thus matching the reported size and zeta potential of typical circulating sEVs [23]. Dynamic light scattering analysis identified a homogeneous population, which correlated to electron microscopy measurements, and a production rate of 2.7 ± 0.3 sEVs per cell (range 2.4–3.0) in a 24-h time period (see Materials and Methods for details).

M14-LUC-derived sEVs were also analyzed for the presence of specific EV markers by fluorescence activated cell sorter (FACS). sEVs were immunocaptured using magnetic beads conjugated with an antibody targeting the human CD63 tetraspanin. We first verified that sEVs were really bound to the beads. In Figure 2C, a magnetic bead with the sEVs bound and stained by the Fuse-IT (ibidi) membrane-specific dye is visualized by confocal microscopy. We then verified, by TEM, the morphology of the bead-bound stained sEVs after their detachment from the beads (free sEVs and right panels). FACS analysis of the immunocaptured sEVs evidenced the expression of the sEV surface markers CD81 and CD9 tetraspanin. In addition, we also found significant levels of the tight junction transcription factor ZO-1-associated nucleic acid-binding protein (ZONAB) and the intermediate filament glial fibrillary acidic protein (GFAP) (Figure 2D).

Transmission electron microscopy (TEM) analysis of the sEVs isolated from mouse plasma revealed the peculiar ultrastructure of sEVs (Figure 3A). Immunocapturing of the purified sEVs by magnetic beads conjugated with the anti-human CD63 and further incubation of the bead-bound sEVs with an anti-human CD-81 antibody allowed us to identify the presence of 10% human cell derived sEVs in the mouse plasma (Figure 3B). To be sure that the anti-human CD-81 antibody was specific to the human CD-81 molecule, we tested this antibody on cells of human and murine origin, confirming its specificity for a human epitope (Appendix A).

Regarding circulating miRNA sequencing experiments, a bioinformatics primary analysis of sequence files generated three datasets for each time point: (i) human-specific miRNAs, unique to Homo sapiens species or whose sequences were distinguishable from those of the mouse orthologue miRNAs; (ii) mouse-specific miRNAs, unique to Mus musculus or whose sequences were distinguishable from those of the human orthologue miRNAs; (iii) evolutionary conserved miRNAs, whose sequences were indistinguishable between human and mouse orthologue miRNAs. Normalized read counts (counts per million (CPM)) obtained for the miRNAs of the three datasets found in mouse total plasma as well as enriched in sEVs are provided in Appendix A, respectively.

We focused our attention on the human-specific miRNAs. The hierarchical clustering analysis performed on the total plasma human-specific miRNA dataset evidenced two main clusters distinguishing, within one group, the control samples (miRNAs from total plasma of mice without tumor) together with those at day 7 after tumor intracranial injection and, in the other group the samples at day 14 and 23. The replicates for each time point well clustered together (Figure 4A). The total number of human-specific miRNAs was shown to increase as a function of time after tumor injection, thus indicating that, as the tumor grew in the brain, the accumulation of human-specific miRNAs significantly increased in the mouse blood circulation from day 7 up to day 23 (Figure 4B).

To compare the miRNA profile of the total plasma and sEV cargo, we performed a hierarchical clustering analysis considering the expression data at day 23 post-implantation. We clearly identified two major clusters, distinguishing the total plasma miRNA samples from the sEV-enriched miRNA samples (Appendix A). The differences in the expression levels of the indicated miRNAs between the two clusters were evident, as indicated by a Pearson’s correlation coefficient of 0.0021 (Appendix A).

We then considered the most expressed miRNAs in the two compartments (total plasma and sEV) as percentage with respect to the total expressed miRNAs. After plotting the rank abundance curves for the total and sEV-enriched human-specific miRNAs (Figure 5A), we chose a threshold of 2% expression since an expression lower than 2% was supposedly not relevant for a microRNA function [24]. Thus, we found that only 10/59 and 9/35 human-specific miRNAs have an expression level >2% in the total plasma and sEV-enriched compartment, respectively (Figure 5B). miR-193b-3p is the most abundant miRNA in the total plasma (19.5%), it is absent in the sEV cargo, and is described as a tumor suppressor. Only miR-660-5p and miR-1246 are expressed both in the total plasma (9.7 and 7.1%, respectively) and in the sEV cargo (2.2 and 3.2%, respectively). The most abundant miRNAs in the sEVs are miR-6131, miR-6853-5p and miR-1268a (25.6, 22.8 and 18.1%, respectively).

To investigate the biological functions of the most abundant human-specific miRNAs, we identified their experimentally validated target genes by using DIANA TOOLS TarBase (http://diana.imis.athena-innovation.gr/DianaTools/index.php?r=tarbase/index) and then analyzed their functional annotation with the Database for Annotation, Visualization and Integrated Discovery (DAVID) gene enrichment tool (https://david.ncifcrf.gov/home.jsp). We found that, in both total plasma and sEV compartments, these target genes enrich many Kyoto Encyclopedia of Genes and Genomes (KEGG) pathways all related to cancer, among which is the KEGG term melanoma (Appendix A). These data further demonstrate that those miRNAs released in the blood circulation are derived from the human melanoma cells injected in the mouse brain.

The hierarchical clustering analysis performed on the conserved miRNA dataset did not distinguish any cluster, suggesting that this class of miRNAs pre-exist at relevant levels in the mouse organism, therefore not allowing for an accurate analysis of their possible changes. Nonetheless, we decided to further analyze only those conserved miRNAs that were upregulated during tumor growth (considering the increase over the control mice), assuming that such miRNAs were likely released by tumor cells. After plotting the rank abundance curves for the total and sEV-enriched conserved miRNAs (Figure 6A), we found that most of the conserved miRNAs have an expression level <2%. Only 11/67 and 16/200 conserved miRNAs have an expression level >2% in the total plasma and sEV-enriched compartment, respectively (Figure 6B). Six out of 11 miRNAs detected in the total plasma were also enriched in the sEVs. The most abundant miRNA in the total plasma is miR-16-5p (19.1%), which is also enriched in the sEVs (2.8%). The most abundant miRNAs in the sEV compartment are miR-150-5p, miR-29a-3p and miR-21a-5p. DAVID gene enrichment analysis for the target genes of the most abundant conserved miRNAs identified by DIANA TOOLS TarBase showed that, in both the total and sEV compartment, the enriched KEGG pathways are mostly related to cancer (Appendix A).

In total, nine out of 14 miRNAs secreted in the sEVs released in the culture medium by the M14 cells corresponded to miRNAs found in the sEVs released in the mouse plasma by the M14 injected in the mouse brain, indicating that the in vitro model only partially reflects what happens in vivo (Appendix A).

We also assessed, by quantitative Reverse Transcription-Polymerase Chain Reaction (qRT-PCR), the expression of total plasma and sEV-enriched miRNAs passing the expression threshold of 2% (see Figure 5B and Figure 6B) in mouse brain tumor xenografts at day 23 after intracranial M14 cell injection. Among the 25 most abundant miRNAs secreted in sEVs, 15 (60%) were detected at variable expression levels in brain xenografts (Appendix A), while 17 (68%) out of the 25 most abundant miRNAs released in total plasma resulted expressed at different levels in brain xenografts (Appendix A).

To validate the sEV-associated miRNAs found in our preclinical experimental model in human clinical cases, we performed an extensive literature search for public datasets of EV miRNA profiles in metastatic melanoma patients. To date, we found only one recently published paper that describes a comprehensive microarray profiling of EV-associated miRNAs in melanoma cases at different stages, including eight Stage IV metastatic patients [25]. Starting from original data deposited in the GEO repository (https://www.ncbi.nlm.nih.gov/geo/browse/) (GSE100508), we found that, among the 25 most abundant miRNAs in our sEV-enriched compartment (i.e., nine human-specific and 16 conserved miRNAs with an expression level >2%, see Figure 5B and Figure 6B), 20 (80%) were also expressed in EVs from Stage IV metastatic melanoma cases (normalized probe intensity value >5 in at least 50% of samples) (Figure 7A). Moreover, all but one (miR-224-5p) of these 20 miRNAs were also present in EVs of healthy controls included in the same public dataset.

Moreover, to investigate circulating EV-miRNAs potentially associated with disease progression, we re-analyzed the public dataset by comparing metastatic (*n* = 8 Stage IV) to primary (*n* = 2 Stage II) melanoma cases. Noticeably, even though they did not reach statistical significance (probably due to the small sample size), three out of the 20 expressed miRNAs, including one human-specific (miR-224-5p) and two conserved miRNAs (miR-130a-3p, miR-21-5p) had a trend of upregulation during melanoma progression, from healthy controls to primary melanoma and metastatic cases (Figure 7B).

## 3. Discussion

Metastatic melanoma has the highest risk of spreading to the central nervous system among the most common cancers, as up to a quarter of patients have brain metastases at diagnosis, with very limited treatment options and poor prognosis [1,2,3,5]. Metastatic melanoma lacks robust prognostic biomarkers, which could translate into adequate therapeutic interventions and survival benefits for patients. Nonetheless, several interesting reports have tried to define the role of circulating miRNAs in melanoma progression, even though the different detection and normalization methods used prevent the identification of a definite miRNA signature for the early detection of melanoma [26,27,28,29].

In this study, we developed an orthotopic melanoma xenograft model to see if it was possible to simulate the release of sEV in the circulation, in a similar way to human melanoma brain metastasis. For this reason, the melanoma cell line used for our experimental model was chosen among those that recapitulate most of the characteristics common to melanoma brain metastasis. In fact, they harbor a mutation in BRAF V600E/K and CDKN2A genes (see Cosmic website, https://cancer.sanger.ac.uk/cosmic); they have low expression of PTEN [30]; they have an activated PI3K/Akt pathway [31] evidenced by the high expression of genes induced by this pathway, such as PDK1, MTOR, EIF4EBP1, RPS6KB1, MYC, CCND1, SGK1 RICTOR ([32] for gene expression in M14 cells, see also the CellExpress gene expression database for cancer cell lines and clinical samples, http://cellexpress.cgm.ntu.edu.tw/; [33]). In addition, melanocytes arise from the same embryonic germ layer as the brain, so the brain might be a natural environment for melanoma tumors to grow and promote tumor progression [34].

We identified a small set of miRNAs enriched in sEVs released in the blood circulation unequivocally by melanoma tumor cells implanted in the brain. The identification of miRNAs associated with EVs that are specifically released by cancer cells is extremely challenging. Although EVs are commonly released by most cell types of the human body, until now, few specific markers associated with EVs released by tumor cells have been established [35]. Our experimental xenograft model proved to be valuable to identify such miRNAs. Indeed, out of a total 420 miRNAs detected by sequencing the total mouse plasma, we found 50 different human-specific miRNAs (12%) whose expression increased in the plasma as the melanoma tumor grew in the brain, starting from cell injection and up to 23 days. Most of these human-specific miRNAs found in the total plasma are represented by miRNAs with documented tumor suppressor functions, whose secretion is selected during tumor progression as a mechanism to coordinate the metastatic cascade that increases cancer cell survival [36,37,38,39,40,41,42,43,44,45,46].

We focused on the circulating miRNAs contained in the EVs, and, in particular, in the sEVs, which have been demonstrated to play a crucial role in the cell-to-cell communication of tumor cells [47,48,49,50,51].

On the sEV surface, we found the expression of known vesicle markers, such as tetraspanin CD81 and CD9. Moreover, for the first time, we found the presence of two molecules on the external sEV surface: the transcription factor ZO-1-associated nucleic acid-binding protein (ZONAB) [52,53], known to interact with gap junctions, and the glial fibrillary acidic protein (GFAP), which is a specific marker of the glial cells. Indeed, proteomic data are present in the literature, which report the presence of these two molecules in the EVs [54,55] The presence of ZONAB on the surface of sEVs could be involved in their docking onto the membrane of recipient cells. On the other hand, it was demonstrated that miRNAs could be transferred from one cell to another through the gap junctions [56,57]. The presence of GFAP on melanoma cells has been described in the literature [58,59] and could be explained by the shared embryo origin of melanocytes and brain cells [34].

The 10% of human CD81 positivity detected in sEVs purified from the mouse plasma suggests that this could be the fraction of human sEVs released in the mouse bloodstream by the melanoma tumor cells in the brain. This is consistent with the fraction of human-specific miRNAs (12%) we sequenced in the mouse total plasma, further confirming the worth of our experimental xenograft model to study EVs specifically released by cancer cells.

Very little is known about two of the most expressed miRNAs we found in the sEV cargo (miR-6131 and miR-6853) [60]. Understanding the novel functions associated with sEV-enriched miRNAs is beyond the scope of this paper. However, these results warrant further investigation addressing the specific functions of these two rarely characterized miRNAs. Interestingly, unlike the miRNAs identified in the total plasma, which are mostly tumor suppressors, many miRNAs contained in the sEVs have previously shown oncogenic functions (miR-1268a, miR-4258, miR-1246, miR-660, miR373) [61,62,63,64,65,66]. Among the multiple strategies developed by cancer cells to gain a survival advantage, sEVs are, indeed, one of the most concealed. Various studies in the literature described the capacity of cancer cell-derived sEVs not only to circumvent the canonical cellular defenses, but also to reshape the microenvironment of cells towards a malignant phenotype. By surrounding targeted tissues with pro-tumorigenic signals, cancer cell-derived sEVs can help in priming the pre-metastatic niche [67,68,69,70]. Only one miRNA with tumor suppressive functions (miR-122-3p) was found to be associated with sEVs [71]. As discussed above, tumor cells may release such miRNAs in the extracellular space to increase their ability to survive.

The analysis of the miRNAs contained in the sEVs secreted by our xenografted M14 melanoma cell line revealed only a partial concordance among the most abundant miRNAs found in the cell culture and those found in the sEVs purified from the mouse plasma, with nine out of the 14 (about 60%) being consistently highly expressed in both specimens. Indeed, it is not surprising that cells in culture behave differently from the same cells injected as xenografts in mice. According to other authors [72], miRNAs expressed in isolated tumor cells growing in vitro only partially correspond to those expressed in tumors growing in animals. Indeed, in vivo, miRNAs are influenced by external stimuli in the tumor microenvironment such as the acidic extracellular environment, the presence of hormones [73] or cytokines [74], external physical stimuli [75] or hypoxia [76].

The analysis of the miRNAs expressed in the tumor xenografts excised from the mouse brain at the time of sacrifice revealed that not all the miRNAs secreted in the circulation were previously accumulated in the donor cells. It can be hypothesized that miRNAs secreted into sEVs can be either accumulated in parental cells, to be released as needed to target cells, or they can be released without prior accumulation [77].

The most significant KEGG pathway targeted and possibly downregulated by the most abundant sEV-contained miRNAs included several genes related to cancer progression such as NRAS. We hypothesize that the downregulation of NRAS in a sEV-recipient cell is consistent with a survival signal promoted by the M14 sEV-donor cells, which harbor an activating BRAF V600E/K mutation (see Cosmic website, https://cancer.sanger.ac.uk/cosmic), known to constitutively activate the downstream MEK/MAPK kinase signaling pathway. In addition, NRAS mutations cause the constitutive activation of the MEK/MAPK signaling pathway. The double gene alteration could lead to the over-activation of the RAS/RAF/mitogen activated protein kinase-signaling pathway, promoting senescence [78,79]. Our idea is that sEVs released by tumor cells with the BRAF mutation harbor messages to avoid such “oncogene antagonism” by targeting NRAS in the recipient cells, which potentially harbor NRAS mutations. Another interesting miRNA target gene is PDGFRA (a family of cell surface type III receptor tyrosine kinases), whose activation also initiates downstream signaling cascades such as MAPK, possibly functioning as a BRAF competitor gene as well. More intuitive is the targeting of genes which can be considered as tumor suppressors, such as HOXD10, which has been demonstrated to negatively regulate neoplasm metastasis by downregulating the expression of MMP2 and MMP9 [80], TP63, since its loss of expression is causative of increased progression and metastasis in vivo [81], RECK, which is an inhibitor of matrix metalloproteinase [82], the well-known Cyclin-Dependent Kinase Inhibitor 1B (CDKN1B) [83] and serine/threonine-protein kinase 4 (STK4), which has been identified as a tumor suppressor in many cancer types [84]. Moreover, the inhibition of E2F2 can contribute to increased tumorigenesis [85,86]. Interestingly, most of the target genes discussed above are targeted by the human-specific miR-224-5p, with the exception of TP63, targeted by miR-1246, and STK4, targeted by miR-373-3p. miR-224-5p was initially classified as a tumor suppressor [87], but it is emerging that miR-224-5p could also display oncogenic functions [88,89,90]. Very interestingly, we found that this miRNA is more expressed in the EVs from metastatic melanoma patients compared to primary cases and is absent in the EVs from healthy controls, thus being a potentially interesting biomarker of melanoma progression.

Regarding the most abundant sEV-enriched human-specific miRNA, namely miR-6131, we noticed that some of its predicted target genes are the same as the experimentally validated targets of other sEV-enriched miRNAs (e.g., RAS, PDGFR and E2F2). In addition, they enrich some of the same KEGG pathways, such as “Glioma” and “Melanoma”, thus suggesting a new possible crucial role of this quite unknown miRNA in the progression of melanoma tumors. The target gene prediction analysis of the second and fourth most abundant miRNAs, i.e., miR-6853 and miR-4258, only gave a limited number of predicted target genes for both of them (less than 40), which did not cluster in any defined pathway. miR-1268a instead gave validated target genes, which are included in the gene ontology analysis presented in Appendix A.

A separate discussion is needed for the evolutionarily conserved miRNAs, those miRNAs that are indistinguishable between humans and mice. Since they are already highly expressed in the mouse sEVs, we hypothesized that only those that increased their expression during tumor growth are probably released by human tumor cells. Even though the presence of these miRNAs in the mouse sEVs flattens their expression levels, we identified 16 conserved miRNAs whose expression increases in the plasma as the tumor grows. Among them, oncogenic miRNAs were not exclusively enriched in the sEVs but were detected in both the total plasma and the vesicles.

Twenty out of the 25 most abundant miRNAs we found in the EVs released by melanoma cells in the mouse bloodstream were validated in silico, in serum EVs from a cohort of metastatic melanoma cases included in a public dataset [25]. Remarkably, a small set of these sEV-associated miRNAs, namely miR-224-5p, miR-130a-3p and miR-21-5p, showed a trend of upregulation during tumor progression from healthy controls to primary ones, as well as metastatic melanoma cases, and thus could be potential candidate biomarkers for melanoma progression. Concordantly, miR-21 (*p* = 0.038), was found to be expressed by qPCR at higher levels in the plasma-derived exosomes of metastatic melanoma patients compared to unaffected controls [91]. In addition, when using the TCGA (The Cancer Genome Atlas) data for 216 independent melanoma cases, the miR-21 expression level was shown to increase in tumor tissues according to patients’ Clark level, associated with their melanoma tumor grade [91], as already reported for miRNAs with multiple targets with both oncogenic and tumor suppressor functions [92,93,94]. Furthermore, miR-224-5p could be one of those miRNAs with a dual function as both a tumor suppressor and an oncogene. Our data are consistent with this new definition for miR-224-5p. miR-130a was found to be increased in cancer tissues and serum samples of colon cancer patients and is proposed as a possible marker for early tumor diagnosis [95].

## 4. Materials and Methods

### 4.1. Cell Line Maintenance

The M14 human metastatic melanoma was chosen for the cellular model, as this line recapitulates most of the characteristics of melanoma cells that metastasize to the brain [96]. The M14 melanoma cell line was transfected with a vector pcDNA3-luc containing the firefly luciferase gene and underwent antibiotic selection for two weeks. M14-LUC cells were cultured in RPMI-1640 medium supplemented with 10% fetal calf serum (Euroclone, Pero, Italy), L-glutamine (1%) and antibiotics at 37 °C in a 5% CO_2_ air atmosphere in a humidified incubator.

The identity of the M14 cell line was confirmed and certified by analyzing the genetic characteristics of the cell line by PCR (polymerase chain reaction) single-locus technology. In total, 21 independent PCR systems, Amelogenin, D3S1358, D1S1656, D6S1043, D13S317, Penta E, D16S539, D18S51, D2S1338, CSF1PO, Penta D, TH01, vWA, D21S11, D7S820, D5S818, TPOX, D8S1179, D12S391, D19S433 and FGA were investigated using the Promega PowerPlex 21 PCR Kit (Promega Corporation, Madison, USA). In parallel, positive and negative controls were carried out, yielding correct results. The genetic results were then compared with the online database of the DSMZ (German Collection of Microorganisms and Cell Cultures, GmBH, https://www.dsmz.de/). The Eurofins Medigenomix, Forensik GmbH Company (Ebersberg, Germany) performed the analysis.

### 4.2. In Vivo Intracranial Xenografts

Institute for Cancer Research, Caesarean Derived-1 [ICR (CD-1)] outbred athymic (nu/nu) male mice, 6–8 weeks old and weighing 22–24 g, were purchased from Charles River Laboratories (Calco, Italy). All procedures involving animals and their care and protection were in compliance with the national and international directives (D.L. 4 March 2014, no. 26; directive 2010/63/EU of the European Parliament and of the council).

Mice were anesthetized and injected in the brain with M14-LUC cells at 2.5 × 105 cells/mouse, through the central-middle area of the frontal bone to a 2 mm depth, using a 0.1 mL glass microsyringe and a 27 gauge disposable needle, as previously described [97]. One hour prior to intracranial implantation, mice were weighed and pre-medicated with an oral administration of 0.5 mg/kg of Metacam (meloxicam) to control for post-operative pain and inflammation. The medication was administered until the end of the experiment. Animals were closely monitored by visual inspection and weighed daily.

Tumor growth was monitored weekly using the IVIS imaging system 200 series (Caliper Life Sciences, Hopkinton, MA, USA). Briefly, mice were anesthetized with a combination of tiletamine-zolazepam (Telazol, Virbac, Carros, France; 50 mg/kg) and xylazine (xylazine/Rompun, Bayer Ltd. Animal Health Div, Isando, South Africa; 7.5 mg/kg) given intraperitoneally. Then, mice were injected with 150 mg/kg D-luciferin (Caliper Life Sciences), and imaged 10 min after luciferin injection. Imaging was performed at days 7, 14 or 23 after tumor cell injection. Data were acquired and analyzed using the living image software version 3.0 (Caliper Life Sciences).

For an analysis of the total plasma-circulating miRNAs, about 100 µL/animal of blood was collected from the caudal vein into light blue vacutainer tubes, before tumor cell injection and at days 7, 14 or 23 after tumor cell injection. Plasma was immediately prepared by centrifugation at 1200 rpm and stored at −80 °C. At day 24, mice were euthanized, and brains were excised for histological analysis.

For the analysis of sEV-contained miRNAs, about 1000 µL/animal of blood was collected into light blue vacutainer tubes by cardiac puncture at day 23 after tumor cell injection. Plasma was immediately prepared as described above and stored at −80 °C. The mice were immediately sacrificed after blood collection and their brains were excised to perform a subsequent histochemical analysis.

### 4.3. Ethical Review Procedure

All animal studies and procedures were undertaken at the Institute of Biochemistry and Cell Biology, National Research Council/Infrafrontier (Monterotondo, Italy). Experiments were performed in accordance with general guidelines regarding animal experimentation, after being reviewed by the established local Animal Welfare and Ethical Review Body (AWERB), and approved by the Italian Ministry of Health (Refs 67/2016-PR), as the competent authority, in compliance with the Legislative Decree 26/2014, ref.13/2010-B and with the national legislation on animal welfare, transposing the 2010/63/EU directives on the protection of animals used in research.

Experimental mice that underwent intracranial injection were accurately inspected and monitored daily (during the 23 days of the study) and we detected mild and moderate weight loss (less than 5%) and minor dehydration. No major clinical symptoms were revealed, including stress, pain and discomfort, which would have been considered incompatible with animal health and welfare.

### 4.4. Histochemistry

The excised brains were fixed with 4% phosphate buffered formalin, and 4 μm paraffin-embedded sections were stained using hematoxylin and eosin (H&E). The sections were then subjected to morphological analysis by an optical microscope at 4× magnification. Sagittal brain sections were observed in a Bright Field using a motorized Widefield scope (LMD 7000, Leica Microsystems, Buccinasco, Italy). For each section, the entire area was automatically acquired in xy and a tiled image was generated by the software (LAS X, Leica Microsystem).

### 4.5. sEV Isolation and Purification from Cell Cultures

M14-LUC was seeded in RPMI (Gibco, Thermo Fisher Scientific, Waltham, MA, USA) and 10% Fetal Bovine Serum (FBS; Hyclone, Cytiva, Marlborough, MA, USA) to obtain 80% confluence after 48 h of cell growth. Cell plates were then washed twice with PBS (Gibco) and incubated for 24 h in RPMI supplemented with 10% EV-free FBS at 37 °C and 5% CO2. FBS used for sEV production was depleted from endogenous EVs prior to use by ultracentrifugation at 100,000× *g* for at least 8 h, using an L8-70MK ultracentrifuge (Beckmann Coulter, Pasadena, CA, USA). After centrifugation, the FBS supernatants were filtered with a 0.22 µm filter (ThermoFisher Scientific, Waltham, MA, USA) and stored in aliquots at −80 °C.

Small EVs were prepared from the cell culture media using differential centrifugation steps. All preparation and centrifugation steps were performed at 4 °C. The collected cell media were subjected to a first centrifugation at 300× *g* for 10 min to remove non-attached cells, followed by a second centrifugation at 2000× *g* for 30 min to remove apoptotic bodies and, finally, the supernatant was filtered through a 0.22 µm-filter to remove residual cell organelles and larger microvesicles. sEVs were then pelleted from the purified supernatant by ultracentrifugation at 100,000× *g* for 70 min in 38 mL polycarbonate tubes (Beckman Coulter, Brea, CA, USA). This sEV-enriched pellet was dissolved in PBS and the ultracentrifugation was repeated again, as above. The final sEV pellets were then used for subsequent applications.

### 4.6. sEV Isolation and Purification from Blood Plasma

Plasma obtained as described above was pooled from nine mice and diluted at 50% with PBS. All preparation and centrifugation steps were performed at 4 °C. Plasma samples were subjected to a first centrifugation at 2000× *g* for 30 min, followed by a second centrifugation at 12,000× *g* for 45 min. sEVs were then pelleted from the purified plasma by ultracentrifugation at 100,000× *g* for 2 h. This sEV-enriched pellet was dissolved in PBS and the ultracentrifugation was repeated, as above. The final sEV pellets were then used for subsequent applications.

### 4.7. Dynamic Light Scattering (DLS) and Zeta Potential

All the measurements were made at 25 °C on a Zetasizer Nano ZS spectrometer (Malvern Panalytical, Malvern, UK) equipped with a 5 mW HeNe laser (wavelength λ = 632.8 nm) and a non-invasive backscattering optical setup. Solvent-resistant micro cuvettes (ZEN0040, Malvern, Herrenberg, Germany) were used for experiments with a sample volume of 40 µL. The results are given as the mean ± standard deviation of ten replicates. sEV size distributions and concentrations were calculated by a recently proposed DLS-based non-invasive tool. Briefly, DLS was used to measure the intensity distribution of each sample. sEVs were approximated to core-shell spherical lipid bilayer vesicles to obtain a form factor. The refractive index of the sEV shell was set as equal to that of the plasma membrane (nL = 1.46) and the refractive index of cytoplasm (nC = 1.38) was used for the exosome core. The Rayleigh Ratio (RR) of the samples was calculated by obtaining the intensity of light scattered by each sample and by the buffer solution. When the area of the intensity distribution obtained with DLS is set as equal to the RR, and the number distribution of the sample is calculated using the form factor, the concentration of sEVs per ml can be calculated. More details can be found in Palmieri et al. [98]. The zeta potential of samples was calculated with a Zetasizer Nano ZS (Malvern Panalytical). To obtain the number of vesicles per cell, the DLS data have been divided by the cell number. The Zeta potential was calculated from the electrophoretic mobility by means of the Henry correction to Smoluchowski’s equation [99].

### 4.8. Light and Electron Microscopy

For ultrastructural analysis by transmission electron microscopy (TEM), pellets containing purified exosomes were fixed with a mixture of 4% (*w*/*v*) paraformaldehyde (PFA), 1% (*w*/*v*) glutaraldehyde (TAAB, Aldermaston, UK) in 0.1 M sodium phosphate buffer pH 7.1. Subsequently, 5 μL of exosome-containing solution was placed on formvar-carbon-coated 100 mesh grids (Electron Microscopy Science, Hatfield, PA, USA) for 20 min at room temperature. Samples were then fixed with 1% (*w*/*v*) glutaraldehyde (TAAB) in pure water, rinsed with pure water and counter-stained with a mixture of oxalacetate/uranyl aetate pH 7 followed by uranyl aetate/methyl cellulose pH 4. Upon drying, grids were observed with a Jeol 1010 transmission electron microscope (TEM; Jeol, Peabody, MA, USA). Images were obtained using a Gatan MSC 791 CCD camera (Gatan, Pleasanton, CA, USA).

For confocal microscopy, purified exosomes were immunocaptured with magnetic beads (see Flow Cytometry) and labelled either with the membrane-specific Fuse-It dye (ibidi) or the anti CD81-PE antibody. Then, they were spotted on glass and mounted with a 0.17 mm-thick coverglass. Samples were observed with a TCS SP5 laser scanning confocal microscope, (Leica Microsystems, Mannheim, Germany) using a Plan Apochromat 63X (numerical aperture = 1.45) oil immersion lens with an optical pinhole at 1AU. An Argon multiline laser operating at 488 nm, and a HeNe at 543 nm were used as excitation sources. Confocal Z-stacks were collected with 0.13 µm intervals with a 2 µm total optical depth.

Images for direct comparison were collected under the same parameters and representative images were chosen from multiple assays.

### 4.9. Flow Cytometry

FACS characterization of sEVs purified from cell culture medium was performed by previously immunocapturing of the vesicles with magnetic beads of 4 µm in diameter conjugated with the CD63 tetraspanin, following manufacturer’s recommendation’s (Dynabeads^®^, Invitrogen, Carlsbad, CA, USA). The bead-bound sEVs were then subjected to indirect or direct immunofluorescence to detect the presence of specific surface markers. Antibodies: anti-hCD81-PE (R&D Systems, Minneapolis, MN, USA), anti-moCD81-PE (R&D Systems), anti-hCD9-FITC (R&D Systems, clone #209306), anti-ZONAB (Sigma-Aldrich, St. Louis, MO, USA), anti-GFAP (Dako Cytomation, Carpinteria, CA, USA).

FACS analysis was also performed on sEVs purified from mouse plasma after the immunocapture of the vesicles by magnetic beads conjugated with the CD63 tetraspanin, as described above. The bead-bound sEVs were then subjected to direct immunofluorescence to evidence the presence on their surface of the specific marker CD81 using the antibody anti-hCD81-PE (R&D Systems).

### 4.10. Total RNA Preparation

To perform an analysis of the total plasma-circulating miRNAs, for each data point, two pools of 200 µL of plasma were obtained from eight mice, pooling 50 µL of plasma from each animal. Plasma was extracted at different times and pooled from the same four animals, matched by balancing the total weight of the tumors between the two pools. Total RNA was extracted from the plasma pools using the Total RNA Purification Kit (Norgen Biotek, Thorold, ON, Canada), according the manufacturer’s instructions.

Total RNA was isolated from the pelleted sEVs purified from the plasma of nine mice, as well as from M14 cell culture media using the Fatty Tissue RNA Purification Kit (Norgen Biotek). Before RNA extraction, the sEV preparations were treated with proteinase K followed by RNAse A to avoid contamination of the internal sEV-cargo with RNA adsorbed on the vesicles’ external surface.

Total RNA was extracted using the Fatty Tissue RNA Purification Kit (Norgen Biotek) from the brain tumor xenografts of eight mice sacrificed at day 23 after intracranial tumor cell injection.

The isolated RNA was analyzed on an Agilent 2100 Bioanalyzer, using the Agilent RNA 6000 Pico kit (Agilent Technologies, Santa Clara, CA, USA).

### 4.11. Plasma and sEV-Carried Total Small RNA Sequencing

Starting from total RNA extracted from total plasma, plasma-purified sEVs and M14 culture media-purified sEVs, small, barcoded RNA libraries were generated using the Applied Biosystems SOLiD™ Small RNA library protocol (July 2011 Rev.B) with the Total RNA-Seq Kit (Applied Biosystems, Foster City, CA, USA). Two independent replicates at 0, 7, 14 and 23 days, each consisting of a pool of four mice, were used for the total plasma; in the case of purified sEVs, two pools of nine mice were analyzed at day 23. The libraries were size selected to enrich the miRNA fraction, then pooled and sequenced on the SOLiD 5500XL platform (Applied Biosystems) at Genomnia Company (Bresso, Milan, Italy).

### 4.12. miRNA Bioinformatics Analysis of Mouse Plasma-Derived Samples

Since the small RNA libraries showed a hybrid composition of human and murine sequences, the analysis of miRNAs contained in the mouse-derived samples was carried out using ad hoc procedures. Sequence files in ColorSpace format (csfasta and qual) included in the xsq file were used to generate fastq files, using the software prinseq-lite. Fastq files were converted into fasta files. Reads in fasta format were aligned using Blastn software (version 2.2.29+, https://blast.ncbi.nlm.nih.gov/Blast.cgi) to the mature microRNA sequences of Homo sapiens and Mus musculus in the mirBase database (version 21, http://www.mirbase.org/). Blastn was executed locally with the following parameters: -task blastn-short-max_target_seqs 1-dust no. Blast output was filtered by proprietary scripts to extract only the perfect alignments (zero mismatches). Finally, the counts of the reads associated to each human or murine microRNA were generated.

To identify pairs of identical miRNAs between humans and mice, a Blastn alignment was performed using, as queries, the mature human miRNA sequences found in miRBase and, as targets, the sequences of mature murine miRNAs. Blastn was executed locally with the following parameters: -task blastn-short -dust no. The output of Blastn was filtered to keep the alignment in which the human miRNA was perfectly aligned along its entire length against a murine miRNA (of equal or greater length) or, vice versa, the murine miRNA was perfectly aligned along its entire length against a human miRNA (of equal or greater length). The identified microRNA pairs were classified as “conserved” miRNAs. These miRNAs were removed from the count files and analyzed separately.

The miRNAs counts were subsequently separately analyzed for differential expression for conserved and human-specific miRNAs. The analysis was performed with the edgeR statistical library of the Bioconductor environment on R at 64-bit (version 3.0.2), which performs count normalization and calculates differential expression statistics. Data were further corrected for miR-451, which was eliminated by counts as it was shown to be exclusively expressed by red blood cells [24,100,101].

### 4.13. Identification of miRNA Target Genes and Functional Enrichment Analysis

Human genes targeted by miRNAs of interest were identified using DIANA TOOLS TarBase (v6.0), which is a manually curated database collecting only experimentally validated targets [102]. The DAVID database was used to perform a functional enrichment analysis of the identified target genes on their KEGG pathway annotation [103]. Annotation clusters were obtained using the Functional Annotation Clustering analysis option, with default parameters.

For the most abundant sEV-enriched human-specific miRNAs (miR-6131, miR-6853-5p, miR-4258), since DIANA TOOLS TarBase does not include them, their target genes were predicted by the miRWalk (v3.0) tool, using TarPmiR, targetScan and miRDB predictions on miRBase 22 [104].

### 4.14. Analysis of EV-Associated miRNAs in Human Metastatic Melanoma Cases

A public dataset of EV-associated miRNA microarray profiles in melanoma clinical cases at different stages and healthy controls was retrieved from the GEO repository (GSE100508). Global quantile-normalized probe intensity values, filtered as described in the corresponding paper [25], were downloaded from GEO Series and used for our visualizations. The Bioconductor GEOquery and limma R packages by GEO2R web tool (https://www.ncbi.nlm.nih.gov/geo/geo2r/) were used to perform a differential expression analysis of miRNAs in the EVs of metastatic (*n* = 8 Stage IV metastatic cases) vs. primary melanoma cases (*n* = 2 Stage II primary cases). Adjusted *p*-values were calculated according to the Benjamini–Hochberg method. Box plot visualizations were performed in R using the ggplot2 package.

### 4.15. Analysis of miRNAs Expressed in Xenograft Tissue of M14 Metastatic Melanoma Cells

Total RNA extracted from the brain tumor xenografts of eight mice at day 23 post-implantation were pooled in two samples (four mice/replicate) and reverse transcribed using TaqMan MicroRNA Reverse Transcription kit (Applied Biosystems). cDNA was pre-amplified using TaqMan PreAmp Master Mix (Applied Biosystems). qRT-PCR was performed on an Applied Biosystem 7900HT thermal cycler using TaqMan Human Array MicroRNA A+B Cards v3.0, including 754 miRNA assays (Applied Biosystems, Thermo Fisher, Waltham, MA, USA), according to the manufacturer’s instructions. Applied Biosystems 7900 SDS software (v2.3) was used for the relative quantification analysis. Only the miRNAs found to be expressed in both replicates were considered. Delta Ct were calculated by normalizing each miRNA to the average of two housekeeping genes (MammU6 and U6 snRNA).

### 4.16. Data Availability

Sequencing data have been submitted to the European Nucleotide Archive (https://www.ebi.ac.uk/ena). The Accession number is PRJEB36207.

## 5. Conclusions

In conclusion, we identified a small set of three circulating miRNAs (miR-224-5p, miR-130a-3p and miR-21-5p) contained in sEVs, specifically released by tumor cells that correlated with the progression of melanoma metastatic disease. We emphasize the usefulness of our orthotopic xenograft model in order to identify miRNAs that are specifically released by cancer cells and can be useful as candidate liquid biomarkers of disease progression to be monitored in melanoma patients’ blood. To the best of our knowledge, this is the first paper using an advanced xenograft model that supports the emerging role of liquid biopsy for the detection of metastatic melanoma in the brain and warrants further investigations in this growing field.

## Figures and Tables

**Figure 1 cancers-12-01635-f001:**
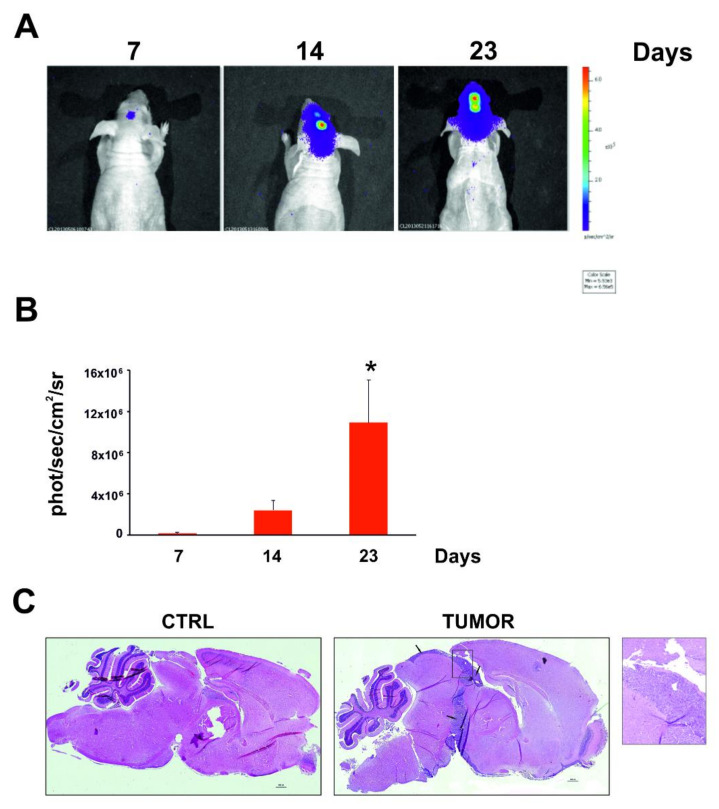
In vivo imaging of M14 metastatic melanoma cells transfected with pcDNA3-luc plasmid (M14-LUC stable cell line) intracranial xenografts. (**A**) Luminescence of M14-LUC cells intracranially injected in the mouse brain was monitored weekly using the In Vivo Imaging Systems (IVIS, 200 series, Caliper Life Sciences, Hopkinton, MA, USA). Mice were anesthetized to perform measurement of luminescence. Representative tumor images on day 7, 14 and 23 after tumor cell injection are shown. Data were acquired and analyzed using the Living Image Software version 3.0 (Caliper Life Sciences). (**B**) Histograms report bioluminescence at day 7, 14 and 23 after tumor cell injection. Error bars indicate ± SD. Unpaired-one-tailed student’s *t* test was used for statistical comparison between the different groups and the control (CTRL) group (* = *p* < 0.05). (**C**) Representative histopathological examination of mouse brains in control (CTRL) and tumor-bearing (TUMOR) mice after 23 days from tumor cell injection. Images shown are at 4× magnification.

**Figure 2 cancers-12-01635-f002:**
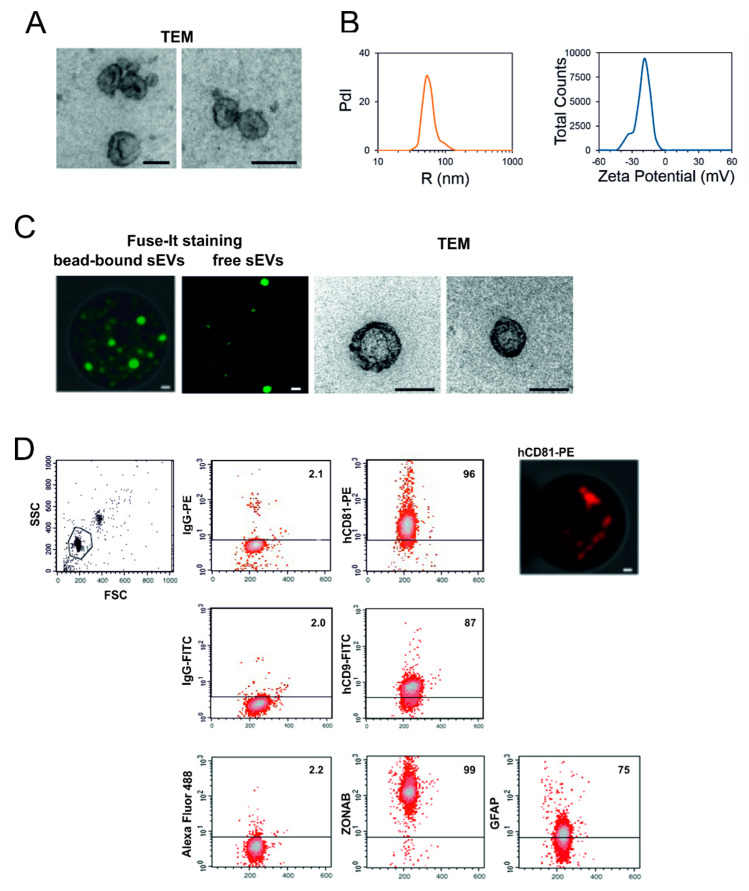
Characterization of M14-released tumor-secreted small extracellular vesicles (sEVs) in cell culture. (**A**) Morphological examination of small extracellular vesicles (sEVs) purified from M14 cell culture medium was performed by transmission electron microscopy (TEM). Bars, 100 nm. (**B**) Size and number of the released sEVs was measured by dynamic light scattering. The representative Intensity distribution curve and Zeta potential distribution are an average of five different measurements of the same sample. (**C**) sEVs purified from cell culture were immunocaptured by magnetic Dynabeads conjugated with CD63 tetraspanin. The bead-bound sEVs stained by Fuse-It membrane-specific dye were studied by confocal microscopy (left panel, bars, 500 nm). The stained sEVs were then detached from the beads and analysed by confocal microscopy (middle panel, bars, 500 nm) and by TEM (right panels, bars, 100 nm). (**D**) Bead-bound sEVs were processed for the detection of the indicated molecules by immunofluorescence and flow cytometry. Aggregates and debris were excluded (gating) from the fluorescence analysis, as shown in the cytogram relative to the light scatter parameters (left panel, top). In each cytogram the number reported represents the percentage of positivity for the indicated molecule. As an example, right top panel reported the confocal microscopy of bead-bound sEVs stained with anti-CD81 antibody conjugated with phycoerythrin (PE). Bar, 500 nm. PdI, intensity distribution; SSC, side scatter; FSC, forward side scatter; FITC, fluorescein isothiocyanate; ZONAB, ZO-1-associated nucleic acid-binding protein; GFAP, glial fibrillary acidic protein.

**Figure 3 cancers-12-01635-f003:**
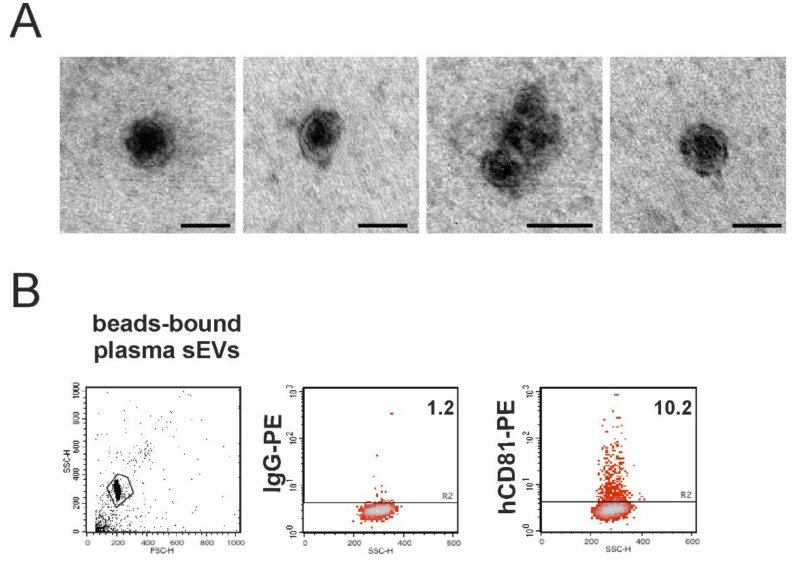
Detection of human sEVs in the mouse plasma. (**A**) TEM analysis of the sEVs purified from the mouse plasma. Bars, 100 nm when not indicated. (**B**) sEVs purified from the mouse plasma were immunocaptured by the magnetic beads conjugated to human CD63 tetraspanin and then processed for immunofluorescence of the human CD81-PE. Aggregates and debris were excluded (gating) from the fluorescence analysis, as shown in the cytogram relative to the light scatter parameters (left panel). The number reported in each cytogram represents the percentage of positivity.

**Figure 4 cancers-12-01635-f004:**
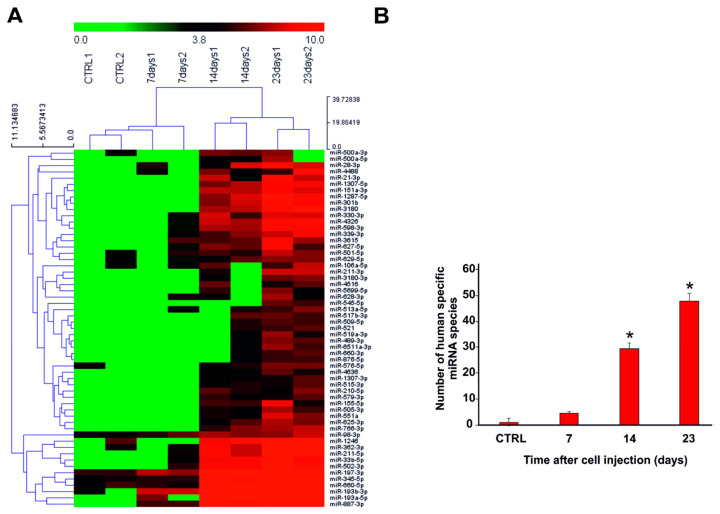
Hierarchical clustering analysis of the human-specific miRNAs released in the mouse total plasma. (**A**) Clustering analysis was performed as a function of the tumor growth in the brain. (**B**) Expression of the human-specific miRNAs released in the total plasma during the tumor growth at different time points after intracranial cell injection. Unpaired-one-tailed student’s *t* test was used for statistical comparison between the different groups and the CTRL group (* = *p* < 0.05).

**Figure 5 cancers-12-01635-f005:**
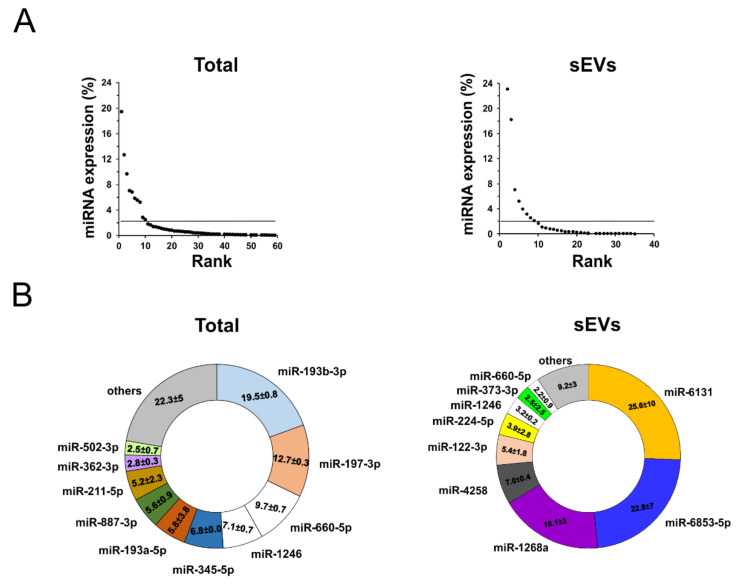
Human-specific miRNA abundance analysis in the mouse plasma. (**A**) Rank abundance curve of the human-specific miRNAs both expressed in the mouse total plasma (left panel) and enriched in the sEVs (right panel). The threshold was set at 2% of the miRNA expression (horizontal line). (**B**) The most abundant miRNAs are shown in the ring graph in both total plasma (left panel) and sEV (right panel) compartments. The number reported in each slice of the ring represents the percentage of expression of the indicated miRNA.

**Figure 6 cancers-12-01635-f006:**
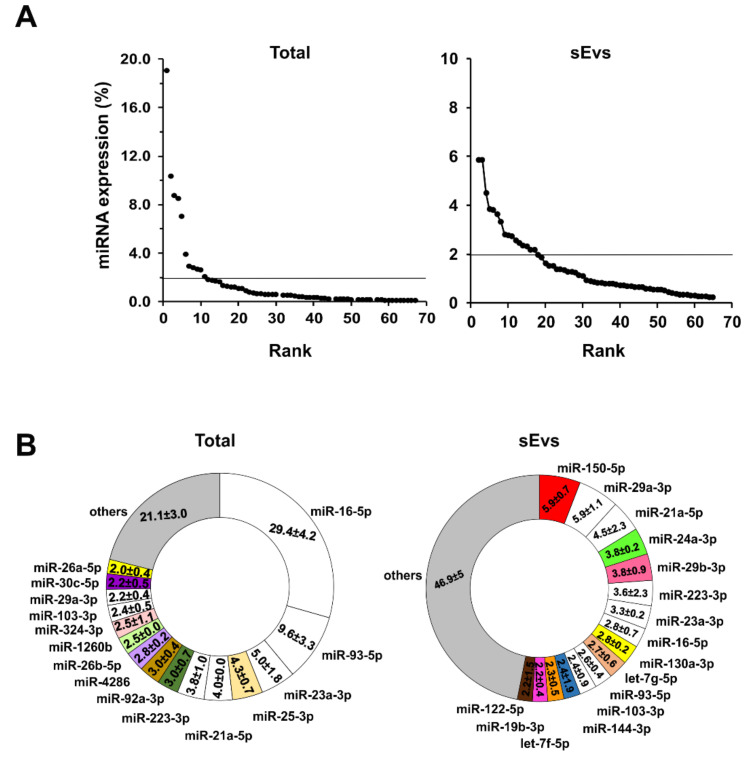
Conserved miRNA abundance analysis in the mouse plasma. (**A**) Rank abundance curve of the conserved miRNAs both expressed in the mouse total plasma (left panel) and enriched in the sEVs (right panel). The threshold was set at 2% of the miRNA expression (horizontal line). (**B**) The most abundant conserved miRNAs are shown in the ring graph in both total plasma (left panel) and sEV (right panel) compartments. The number reported in each slice of the ring represents the percentage of expression of the indicated miRNA.

**Figure 7 cancers-12-01635-f007:**
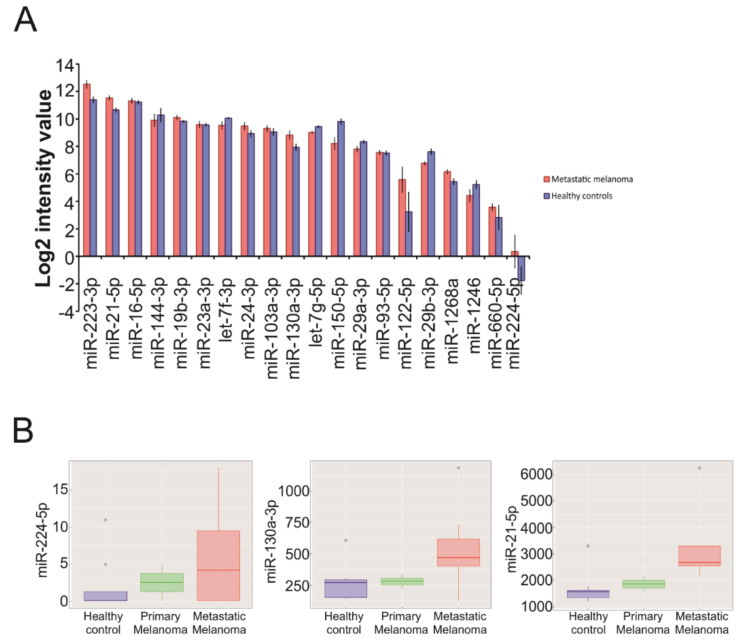
miRNAs expressed in EVs of metastatic melanoma patients included in a public dataset of melanoma clinical cases. (**A**) For the 20 validated miRNAs, normalized expression levels are reported for Stage IV metastatic melanoma group (*n* = 8, red) and healthy controls (*n* = 8, blue) included in the dataset. Values are expressed as mean of the log2 quantile-normalized probe intensity values obtained from GEO Series GSE100508 ± standard error. (**B**) Three EV-associated miRNAs showing a trend of upregulation during melanoma progression from healthy control to primary and metastatic tumors. Boxplots show normalized probe intensity values for each miRNA in healthy controls (*n* = 8, blue), primary (*n* = 2, green) and metastatic (*n* = 8, red) melanoma patients obtained from GEO Series GSE100508.

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
