# Peer review of "Circulating miRNAs in Small Extracellular Vesicles Secreted by a Human Melanoma Xenograft in Mouse Brains"

_cancers, 2020, doi:10.3390/cancers12061635_

Round 1
Reviewer 1 Report
In the proposed manuscript "Circulating miRNAs in small extracellular vesicles secreted by a human melanoma xenograft in mice brain" Guglielmi et.al., try to find melanoma derived EV miRNAs as a liquid biopsie marker for metastasis and tumor growth. The topic of the manuscript is highly interesting but also challanging. The results are well presented but there are doubts regarding the experimental design, results and the interpretation:
major comments:
- Only one cellline was used in this manuscript to identify miRNA candidates. In recent publications comparing miRNA cargo in various melanoma cell line derived EVs it was shown, that miRNA profiles could be very different. Melanoma is known to be very heterogeneous. For this reason one melanoma cell line could not be enough to find melanoma specific EV miRNA markers.
- The authors analysed plasma and EV derived miRNA profiles of human specific and conserved miRNAs. When we compared the results of melanoma EVs from cell culture to the results in the animals we found, that no human specif miRNA, found in the animals, reached the abundance of 2% in the whole cell culture EV miRNA abundance. From this we conclude that the human specific miRNAs are very low abundant at all. So how is the abundance of the human specific miRNAs at all in the melanoma cell line derived EVs? The problem of the conserved miRNAs is, that it is impossible to distinguish between mouse derived and human cell derived miRNAs. Because all cells are expected to secrete EVs the results could be adulterated by mouse cells influenced by the tumor cells. Maybe it would be good to include a miRNA profile of healthy control mice derived EVs / plasma.
- The authors uses a public data set where miRNA profiles of melanoma patient derived EVs where analysed. In this data set also healthy EVs where analysed. How is the miRNA cargo of the reanalysed miRNAs in the healthy EVs?
Questions:
Do the authors have an Idea why there is such a big difference between the plasma and EV derived miRNAs?Which miRNAs (plasma or EV derived) might be better to find specific melanoma markers for liquid biopsies?
How specific is the human CD81 antibody used in flow cytometry to show the 10 % enrichment of human cell derived EVs?
Because of the concerns on the manuscript, we would propose a major revison of the manuscript.
Author Response
- Only one cell line was used in this manuscript to identify miRNA candidates. In recent publications comparing miRNA cargo in various melanoma cell line derived EVs it was shown, that miRNA profiles could be very different. Melanoma is known to be very heterogeneous. For this reason one melanoma cell line could not be enough to find melanoma specific EV miRNA markers.
A: We agree with the reviewer that the use of only one cell line is limiting for a complete human metastatic melanoma miRNA profile and that several cell lines should be investigated. At the same time, many cell lines, which have been cultured for a long time and adapted to in vitro conditions, going through genetic background selection, are not suitable for mimicking the biological and genetic heterogeneity typical of tumors. For this reason, we have chosen a well-characterized and recently established human melanoma cell line that recapitulates as much as possible the melanoma metastatic phenotype in the brain. Therefore, in this work we aimed to test, as a proof of principle, by using an advanced xenograft model of metastatic melanoma in the brain, whether it is possible to identify circulating miRNA packaged in EVs, which are specifically released by tumor cells and whether this model could contribute to progress in this still largely unexplored field of investigation.
- The authors analysed plasma and EV derived miRNA profiles of human specific and conserved miRNAs. When we compared the results of melanoma EVs from cell culture to the results in the animals we found, that no human specific miRNA, found in the animals, reached the abundance of 2% in the whole cell culture EV miRNA abundance. From this we conclude that the human specific miRNAs are very low abundant at all. So how is the abundance of the human specific miRNAs at all in the melanoma cell line derived EVs? The problem of the conserved miRNAs is that it is impossible to distinguish between mouse derived and human cell derived miRNAs. Because all cells are expected to secrete EVs the results could be adulterated by mouse cells influenced by the tumor cells. Maybe it would be good to include a miRNA profile of healthy control mice derived EVs / plasma.
A: Thanks to the reviewer for this comment. We did not find any human specific miRNAs reaching the abundance of 2% in the sEVs purified from the cell culture medium. Apparently, cells in culture behave very differently from the same cells growing as xenograft in mice. This is not surprising if we think to the complexity of in vivo systems comparing to the growth of cells in culture.
In fact, in vivo, miRNAs regulating gene expression and influencing malignant cell behaviour are in turn influenced by external stimuli from the tumor microenvironment, such as acidic extracellular conditions, presence of hormones (1) and cytokines (2), external physical stimuli (3) or hypoxia (4). Our data, clearly highlight the differences between the in vivo and in vitro cell response, supporting the importance of the in vivo xenograft model used in this paper.
We acknowledged in the manuscript, in agreement with the reviewer, the impossibility to distinguish between mouse derived and human cell derived conserved miRNAs. The attempt to clusterize the timecourse data of the whole plasma conserved miRNAs (day 0-control mice, day 7, day 14 and day 23), as we have done with the human specific ones, did not produce clear result, but we were able to identify some conserved miRNAs whose expression level increased during tumor growth (see Fig.xx). We assumed that these upregulated miRNAs were most likely released by the human tumour cells, but we cannot rule out the possibility that these miRNAs could be secreted from mouse cell in response to tumour induction.
New text and relative bibliography have been added in the new version of the manuscript. Lines 355-360 of the revised version.
- The authors uses a public data set where miRNA profiles of melanoma patient derived EVs where analysed. In this data set also healthy EVs where analysed. How is the miRNA cargo of the reanalysed miRNAs in the healthy EVs?
A: Following the reviewer’s suggestion, we analysed the expression levels of the 20 miRNAs of interest, previously validated in the EVs from Stage IV metastatic melanoma cases, in the EVs of the eight healthy controls included in the same public dataset. We found that all but one (miR-224-5p) of these 20 miRNAs were present also in healthy controls and added this new finding in the manuscript (lines 256-257), and in the new Figure 7B.
Questions:
Do the authors have an Idea why there is such a big difference between the plasma and EV derived miRNAs?Which miRNAs (plasma or EV derived) might be better to find specific melanoma markers for liquid biopsies?
A: Thanks to the reviewer for this comment, this is an opened question, which would be solved by the advancement of studies in this field by a better definition of the biological function of EVs in plasma and the associated miRNAs. However, we can hypothesize that in total plasma, the circulating free miRNAs may not derive from an active secretion of intact cells but for example, they could be derived from the lysis of necrotic cells, which have thus released their content into the extracellular space. While, the release of miRNAs inside the EVs is produced by an active secretion, which is regulated by control mechanisms. Furthermore, we preferred to limit our study to only small EVs that seemed to respond better to our initial hypothesis. We do not know what the contents of the larger microvesicles are like. For the reasons above mentioned, we think that sEV might serve as properly reservoirs for tumor biomarkers. In particular, miRNAs contained in these vesicles are stable and surely actively secreted by the cells thus possibly representing a signalling biomarker.
How specific is the human CD81 antibody used in flow cytometry to show the 10 % enrichment of human cell derived EVs?
A: Before using the anti-hCD81 antibody, we tested its specificity on the M14 human cell line and on a cell line of murine origin. The results of this validation have been added in new supplementary Figure S1.
Reviewer 2 Report
The study by Guglielmi et al. explores the miRNA content of small extracellular vesicles (sEVs) released by melanoma cells which were injected intracranially in nude mice. They found a subset of six miRNAs (miR-224-5p, miR-130a-3p, miR-21-5p, miR-23a-3p, miR-29a-3p and miR-29b-3p) which are enriched in tumoral sEVs and correlated with the progression of melanoma. In addition, the authors describe another subset of miRNAs enriched during tumour growth consisting of miR-150-5p, miR-29a-3p, miR-21a-5p. Finally, a public dataset of EV-associated miRNA microarray profiles in melanoma clinical cases at different stage was analysed.
Overall, the topic of secreted biomarkers in liquid biopsies is interesting and the potential that miRNAs and also miRNAs packed in sEVs might have in this respect, is undisputed. Especially miRNAs that might indicate early brain metastasis would be of great clinical relevance. Therefore, the study is of interest, however slightly too preliminary at this stage. Some important control and validation experiments are missing and bioinformatic analysis could and should be extended and improved.
Below are specific points that need to be addressed:
- Lines 130-131 : the authors mention the value of around 2.7 sEVs produced by tumour cell during a period of 24h. A more detailed value is needed (precise average value of the sEVs produced per cell in the treated mice and the sEVs measurements values in the control mice, as well as the error bars in a new figure). Please also mention clearly how the data were obtained in the materials and methods section. In addition, this value should be compared to the production of sEVs produced per M14-LUC melanoma cell in vitro during a period of 24h. This would also help to see if the isolation of sEVs from the plasma was efficient and succesful.
- Figure 3A : please show wider “overview” images for the analysis of sEVs by TEM, only one sEV is shown per picture, which is not conclusive. The EVs are very small (average 52 nm). Pleas add a ref for this statement (line 128, 129).
- The differences between the enrichment scores obtained in Table 1 and Table 2 are not discussed. Both Tables could also move to the Supplements as they are not very informative. Additionally, the whole analysis of miRNA target genes and their potential involvement in specific pathways or functions should be extended to include Targetscan and should be discussed more critically. The target genes are predicted and as we now know, there are many false positive genes in these lists. This could be critically acknowledged in the discussion.
- In this context, Lines 328-350 : the genes found in the enrichment analysis (KECG pathways) are discussed but the specific miRNAs targeting these genes are not specified at all. Please mention which miRNAs targets the mentioned genes.
- A validation by individual qPCRs should be performed in order to confirm the expression level values obtained for the most interesting miRNAs by RNA sequencing. This should be feasible as only few miRNAs were highlighted.
- The authors did not explain the absence of miR-23a-3p, the most abundant miRNA in the sEVs cultivated in vitro but absent of the list of miRNAs obtained from the in vivo A validation of the expression of miR-23a-3p by qPCR should also be done for the sEVs obtained in vitro and in vivo with the orthotopic melanoma xenograft model.
- The way the different clusters were obtained in Table 1 and table 2 is not explained.
- The analysis of EV-associated miRNAs in human metastatic melanoma cases, the number of samples is very low and thus cannot provide significant results. It would have been interesting to compare your data with other public available datasets of miRNA expression detected in sEVS from other melanoma cell lines (metastatic or not). In addition, the choice of miRNAs in this analysis is not clear and is confusing for the reader. MiR-224-5p appears in Figure 5 while miR-130a-3p, miR-21-5p, miR-23a-3p, miR-29b-3p are present in the sEVs values of Figure 6B and miR-29a-3p is shown in the plasma values of Figure 6B.
- Functional assays to explore the potential roles of miRNAs which are enriched in sEVs derived from melanoma cells or which are up-regulated during tumour growth would give the findings more weight.
- The roles of the most abundant human specific miRNAs in sEVs ; miR-6131, miR-6853-5p, miR-1268a, miR-4258, are not discussed and should be tested experimentally. Their target gene are not mentioned.
- The discussion could be shortened and more focused on the purpose of the study, i.e. identification of miRNA biomarkers. The herein found miRNAs should be discussed in more detail in the context of previously published data.
- In this context, some references should be included while others seem out of place or not suitable for the statement. References that could be mentioned: Gowda R. et al., Cancer Treat Rev, 2020; Carpi, S. et al., Exp Rev Mol Diagn. 2020; Cesi, G. et al., Cell Comm Signal 2016; Margue et al., Oncotarget 2015 and others that are covering the miRNA/exosome/biomarker topic.
- Lines 292-294 : The authors should discuss that the presence of ZONAB and GFAP in sEVs was already mentioned in two previous publications, not at the membrane and neither by the same experimental procedure (doi : 10.1074/mcp.M112.022806, doi : 10.1016/j.jprot.2012.12.029, doi : 10.18632/oncotarget.3801, doi: 10.1074/mcp.M900381-MCP200).
- Line 158 onwards: the control sample had no tumor cells injected, so should not have human sequences but the heatmap only shows human miRNAs. This should be better explained or corrected. Overall, a Table would useful to show the numbers of sequence reads for the 3 classes: mouse miRNAs, human miRNAs, both.
- Figure 4: labels are too small.
- Line 266: Why “On the other hand”?
- A good experimental control would be the sequencing of tissue-miRNAs in the brain metastasis of mice (after sacrificing them) to have a comparison between tissue-based miRNAs and secreted miRNAs in plasma.
Author Response
- Lines 130-131 : the authors mention the value of around 2.7 sEVs produced by tumour cell during a period of 24h. A more detailed value is needed (precise average value of the sEVs produced per cell in the treated mice and the sEVs measurements values in the control mice, as well as the error bars in a new figure). Please also mention clearly how the data were obtained in the materials and methods section. In addition, this value should be compared to the production of sEVs produced per M14-LUC melanoma cell in vitro during a period of 24h. This would also help to see if the isolation of sEVs from the plasma was efficient and successful.
A: Following the reviewer’s comment, we detailed how the data were obtained in Materials and Methods section. We also added a more detailed value of the number of sEVs produced by the melanoma M14 cells during a release period of 24 h (line 134).
Concerning the measurement of the sEVs number released in the mice blood comparing control mice and tumor bearing mice we think that this value would not add any reliable information. First, sEVs are secreted by the diverse cells from different organs into blood with likely different secretion rate. Thus, it is very difficult directly distinguishing the presence of sEVs deriving from the tumor only based on their count in the plasma, though performed with a highly sensitive technology such as dynamic light scattering. In addition, a number of factors would affect this value: i) secretion/clearance balance of mouse plasma–derived sEVs, which appear to be measurable only using a specific highly sensitive sEV labelling technology (Akihiro Matsumoto, Yuki Takahashi, Hsin-Yi Chang, Yi-Wen Wu, Aki Yamamoto, Yasushi Ishihama & Yoshinobu Takakura. Blood concentrations of small extracellular vesicles are determined by a balance between abundant secretion and rapid clearance. JEV, 2019, 9:1696517 doi.org/10.1080/20013078.2019.1696517); ii) high heterogeneity between the different mice used in the study; iii) high heterogeneity of tumor growth between the different mice, a very large number of mice should be employed to obtain a reliable value. In our study, we obtained an estimation of the amount of sEVs released in the mice plasma by using a method based on the specific antigen/antibody reaction coupled with fluorescence detection assay. This allowed us to identify a 10% of human (definitely released by the human tumor cells injected in the mice) sEVs on the whole sEV population present in the mouse plasma.
- Figure 3A : please show wider “overview” images for the analysis of sEVs by TEM, only one sEV is shown per picture, which is not conclusive. The EVs are very small (average 52 nm). Pleas add a ref for this statement (line 128, 129).
A: Here the reviewer is suggesting that a larger field of view of TEM images must be shown to be conclusive in our results. However, we have used morphological analysis by TEM just to characterize the ultrastructure and the integrity of purified EVs. The quantitative data on the size and concentration where indeed measured using by dynamic light scattering which is a much more sensitive and accurate method than TEM for vesicle counting. Therefore, we think that replacing the image of few single EVs well showing their ultrastructural morphology with a larger field of view will not add new insights. Just for the reviewer herein we enclose a wider TEM image of the sEVs we analysed.
In addition, we agree with the reviewer that our EVs are small as measured by dynamic light scattering technology. However, it should be noted that the average of 52 nm is not referred to the entire size of the EVs representing instead the average radius of the sEVs analysed, as described in the manuscript line 131.
As requested by the reviewer we added the reference N. 23 for the statement indicated (line 132 of the revised version).
- The differences between the enrichment scores obtained in Table 1 and Table 2 are not discussed. Both Tables could also move to the Supplements as they are not very informative. Additionally, the whole analysis of miRNA target genes and their potential involvement in specific pathways or functions should be extended to include Targetscan and should be discussed more critically. The target genes are predicted and as we now know, there are many false positive genes in these lists. This could be critically acknowledged in the discussion.
A: The enrichment scores reported in the supplementary Table S3 and Table S4 of the revised version are calculated for each annotation cluster by DAVID tool, using the Functional Annotation Clustering analysis option. They are the overall enrichment scores for the annotation clusters based on the EASE score (a modified Fisher Exact P-value) of each term member. The higher, the more enriched.
The enrichment scores shown in Table S4 are higher than those in Table S3 because the target genes used as input list for Table S4 enrichment analysis are much more than target genes used for Table S3 enrichment analysis. Consequently, the p-values of the enrichment of each annotation cluster member are more significant and the global score of the annotation cluster is higher.
To better describe this functional annotation clustering analysis, we added a new paragraph in Materials and Methods section, titled “4.13. Identification of miRNA target genes and functional enrichment analysis” at lines 600 onwards, and added a new reference [N.101 and 102] in Bibliography.
As suggested, we moved the two tables to Supplementary Table S3 and S4, respectively.
As properly reminded by the reviewer, we well know the issues about miRNA target prediction. For this reason, we did not use traditional target prediction tools, such as TargetScan, MiRanda, PITA, but we chose to use DIANA TOOLS TarBase, which is a manually curated database collecting only experimentally validated targets derived from specific, as well as high-throughput experiments, such as microarrays, proteomics, HITS-CLIP and PAR-CLIP (Vergoulis, Nucl Acids Res 2012). To better describe this database, we added a new paragraph in Materials and Methods section, titled “4.13. Identification of miRNA target genes and functional enrichment analysis” at lines 600 onwards, and added a new reference [N.101 and 102] in Bibliography.
- In this context, Lines 328-350 : the genes found in the enrichment analysis (KECG pathways) are discussed but the specific miRNAs targeting these genes are not specified at all. Please mention which miRNAs targets the mentioned genes.
A: Thanks to the reviewer for this comment. We noticed that most of the target genes mentioned in the discussion were targeted by the miR-224-5p. This miRNA was initially classified as tumor suppressor miRNA, but studying more accurately its targets it is emerging that miR-224-5p could also display an oncogenic function. It could be one of those miRNAs with a dual function of tumor suppressor and oncogene. On the other hand, there are already some indication in literature about its possible role as oncomiR (references N. 88-90). Our data also suggest this dual function. Indeed, miR-224-5p targets both genes with tumor suppressor and genes with oncogenic function. Very interestingly we found this gene more expressed in the EVs from the metastatic patients and absent in the healthy samples from the in silico analysis.
All these consideration and new text along with new bibliography have been added in the new version of the manuscript, lines 386-392.
- A validation by individual qPCRs should be performed in order to confirm the expression level values obtained for the most interesting miRNAs by RNA sequencing. This should be feasible as only few miRNAs were highlighted.
A: We do not agree with the reviewer that our RNA-seq data must necessarily be validated by RT-PCR. This is based on the following-reasons. (i) The NGS data regarding levels of gene expression are quite robust and, unlike NGS-identified gene mutations, it is now widely recognized that they must not be necessarily validated with an orthogonal method. (ii) In the case of circulating and EV-carried miRNAs, widely recognized “housekeeping“ miRNAs are not available for use as RT-PCR control reference. (iii) Many of the data presented in this manuscript are based on the differences between the human and the mouse sequence of orthologs miRNAs, and we know that the RT-PCR technology does not always guarantee this possibility.
- The authors did not explain the absence of miR-23a-3p, the most abundant miRNA in the sEVs cultivated in vitro but absent of the list of miRNAs obtained from the in vivo A validation of the expression of miR-23a-3p by qPCR should also be done for the sEVs obtained in vitro and in vivo with the orthotopic melanoma xenograft model.
A: It should be noted that miR-23a-3p, most abundant in the sEVs released by cultured cells (Fig. S3 of the revised version), is among the most abundant conserved miRNAs both in the total plasma and in the EVs (Fig. 6). The class of conserved miRNAs already pre-exist at relevant levels in the mouse organism, not allowing an accurate analysis of their possible changes. Nonetheless, we analysed only those conserved miRNAs that were up-regulated during tumor growth (considering the increase over the control mice), assuming that such miRNAs were likely released by tumor cells and this could be the case of miR-23a-3p. However, it is not surprising that cells in culture behave differently from the same cells injected as xenograft in mice. In vivo, miRNAs regulating gene expression and influencing malignant cell behaviour are in turn influenced by external stimuli from the tumor microenvironment, such as acidic extracellular conditions, presence of hormones, cytokines, external physical stimuli or hypoxia. In the revised version of the manuscript to clarify this point, we added a paragraph, along with relative bibliography, in the discussion, line 355-360 and references N. 72-76.
In addition, miR23a-3p has been also validated in the case series from the public available dataset (Fig. 7A and B). Therefore, we think that qPCR validation is not necessary for the same reasons already explained in the answer to point five. Moreover, the authors think that understanding why some miRNAs are differently expressed between in vitro and in vivo sEVs would require further experiments to be considered in a future work.
- The way the different clusters were obtained in Table 1 and table 2 is not explained.
A: For both Table S3 and Table S4 of the revised version, the functional enrichment analysis was performed by DAVID tool using the Functional Annotation Clustering analysis option. This detail has been added in a new paragraph in Materials and Methods section, titled “4.13. Identification of miRNA target genes and functional enrichment analysis” at lines 600 onwards.
Table S3 includes genes targeted by human-specific miRNAs found expressed in total plasma and sEV (Fig5B-left and right panel, respectively), Table S4 includes genes targeted by conserved miRNAs found expressed in total plasma and sEV (Fig6B-left and right panel, respectively).
- The analysis of EV-associated miRNAs in human metastatic melanoma cases, the number of samples is very low and thus cannot provide significant results. It would have been interesting to compare your data with other public available datasets of miRNA expression detected in sEVS from other melanoma cell lines (metastatic or not). In addition, the choice of miRNAs in this analysis is not clear and is confusing for the reader. MiR-224-5p appears in Figure 5 while miR-130a-3p, miR-21-5p, miR-23a-3p, miR-29b-3p are present in the sEVs values of Figure 6B and miR-29a-3p is shown in the plasma values of Figure 6B.
A: We agree with the reviewer but, at the best of our knowledge, no studies are reported showing miRNA expression in sEV from orthotopic model of metastatic melanoma in the brain, so the comparison of our results with melanoma cells in culture should be misleading in consideration of the cell heterogeneity. At the same time, we would like to outline that, by our model, it is possible to identify candidate circulating miRNA packaged in EVs, which are specifically released by tumor cells and that this model could be very useful to progress in this still largely unexplored field of investigation. These considerations are discussed in the revised manuscript. Moreover, we chose to validate all the miRNAs expressed in sEVs (that is, over 2% of threshold expression level), both human-specific (Fig.5B-right panel, 9 miRNAs) and conserved (Fig.6B-right panel, 16 miRNAs). The selection criteria for the choice of miRNAs of interest is described at line 251 onwords. Regarding miR-29a-3p, it is expressed in total plasma (Fig.6B-left panel), but also in sEVs of our experimental model, as reported in Fig.6B-right panel, and therefore we included it in the validation subset.
- Functional assays to explore the potential roles of miRNAs which are enriched in sEVs derived from melanoma cells or which are up-regulated during tumour growth would give the findings more weight.
A: We agree with the reviewer that experiments on the functionality of the different miRNAs could be of great interest. However, we think that these studies are not the focus of this paper that rather aims to assess how the use of an experimental model of xenograft in vivo can be applied to study sEV-associated miRNAs released by a metastasis at the brain.
- The roles of the most abundant human specific miRNAs in sEVs ; miR-6131, miR-6853-5p, miR-1268a, miR-4258, are not discussed and should be tested experimentally. Their target gene are not mentioned.
A: We agree with the reviewer that the functional role of the most abundant human specific miRNAs in the sEVs should be of interest, but as we already mentioned in the discussion, the understanding of novel functions associated with sEV-enriched miRNAs is beyond the scope of this paper and will be further investigated in a future work. Hence, we did not mention the targets of the two most abundant sEV miRNAs as they are not validated yet, they are only predicted targets.
- The discussion could be shortened and more focused on the purpose of the study, i.e. identification of miRNA biomarkers. The herein found miRNAs should be discussed in more detail in the context of previously published data.
A: The overall discussion has been shortened even though new paragraphs have been added to address the referees’ requests.
- In this context, some references should be included while others seem out of place or not suitable for the statement. References that could be mentioned: Gowda R. et al., Cancer Treat Rev, 2020; Carpi, S. et al., Exp Rev Mol Diagn. 2020; Cesi, G. et al., Cell Comm Signal 2016; Margue et al., Oncotarget 2015 and others that are covering the miRNA/exosome/biomarker topic.
A: We added in the discussion the bibliography suggested by the reviewer. Lines 279-281, references n: 26-29..
- Lines 292-294 : The authors should discuss that the presence of ZONAB and GFAP in sEVs was already mentioned in two previous publications, not at the membrane and neither by the same experimental procedure (doi : 10.1074/mcp.M112.022806, doi : 10.1016/j.jprot.2012.12.029, doi : 10.18632/oncotarget.3801, doi: 10.1074/mcp.M900381-MCP200).
A: As suggested by the reviewer, we discuss now some of the above mentioned papers (lines 323-324) and added them to the reference list (references N. 54, 55).
- Line 158 onwards: the control sample had no tumor cells injected, so should not have human sequences but the heatmap only shows human miRNAs. This should be better explained or corrected. Overall, a Table would useful to show the numbers of sequence reads for the 3 classes: mouse miRNAs, human miRNAs, both.
A: The heatmap reports the actual data obtained from the sequence analysis, without any additional filtering other than the sequence quality score, and notwithstanding, as described in M&M, the reads were directly mapped on miRBase with a mismatch value of 0, the very low signals shown in the control samples by some human specific miRNAs should be accounted to sequence errors.
Following the reviewer’s suggestion, normalized read counts (CPM) obtained for the miRNAs of the three datasets found in mice total plasma as well as enriched in sEVs have been provided in Supplementary Table S1 and S2, respectively (lines 165-167).
- Figure 4: labels are too small.
A: We apologize with the reviewer for this inconvenience probably due to the copy of the figure in the word document of the manuscript. For sharper image, we upload the TIFF document of Fig. 4
- Line 266: Why “On the other hand”?
A: We agree with the reviewer’s comment since it seems more proper to change "On the other hand" to "In addition". Change have been made in the new version line 291.
- A good experimental control would be the sequencing of tissue-miRNAs in the brain metastasis of mice (after sacrificing them) to have a comparison between tissue-based miRNAs and secreted miRNAs in plasma.
A: As suggested by the reviewer, we added in the manuscript new data obtained by miRNA profiling by qRT-PCR, using TaqMan Human Array MicroRNA A+B Cards of M14 melanoma tumor xenografts from mice brain at day 23 after cell injection. Being a technology different from next-generation sequencing, a direct comparison between tissue-based miRNAs and circulating miRNAs found in our experimental model is not feasible. Nonetheless, we can evaluate in these brain tumor xenografts the expression of those total plasma and sEV-enriched miRNAs passing the expression threshold of 2% (those shown in Fig.5B and 6B). These new findings are now described in the manuscript at lines 241-246, and in the new Figure S4.
Round 2
Reviewer 1 Report
Only a minor point: I suggest to include the healthy donor data from the GEO dat set figure 7, in the graphs figure 7c.
If this data also support the message that miR-224-5p,miR-130a-3p, miR-21-5p, miR-23a-3p, miR-29a-3p and miR-29b-3p are upregulated in sEVs during melanoma progression, the manuscript can be published.
Author Response
Only a minor point: I suggest to include the healthy donor data from the GEO dat set figure 7, in the graphs figure 7c.
AR2: As suggested by reviewer, we included the data from healthy donors of the GEO dataset in the boxplot of the new version of Figure 7.
If this data also support the message that miR-224-5p,miR-130a-3p, miR-21-5p, miR-23a-3p, miR-29a-3p and miR-29b-3p are upregulated in sEVs during melanoma progression, the manuscript can be published.
AR2: We are very grateful to the reviewer for his comment. His suggestion helped us to better define the miRNAs associated to the sEVs secreted by the melanoma cells, which could be related to melanoma progression. In fact, by including the healthy donor data together with primary and metastatic cases in the new version of Fig. 7B, we realized that only three out of the previous six miRNAs with a trend of upregulation from primary tumor to metastasis, showed the same trend of upregulation also starting from EVs of healthy donors. Thus, we changed the corresponding paragraphs, including abstract, discussion and conclusions, accordingly and proposed miR-224-5p, miR-130a-3p and miR-21-5p as potential candidate biomarkers of melanoma progression.
Reviewer 2 Report
Some points have not been addressed. See attached document.

Author Response
The points highlighted in yellow should still be addressed.
- Lines 130-131 : the authors mention the value of around 2.7 sEVs produced by tumour cell during a period of 24h. A more detailed value is needed (precise average value of the sEVs produced per cell in the treated mice and the sEVs measurements values in the control mice, as well as the error bars in a new figure). Please also mention clearly how the data were obtained in the materials and methods section. In addition, this value should be compared to the production of sEVs produced per M14-LUC melanoma cell in vitro during a period of 24h. This would also help to see if the isolation of sEVs from the plasma was efficient and successful.
AR2: We apologize to the reviewer because we are unable to produce these data. The new experiments we added in the revised paper derive from experiments we performed before the Covid-19 emergency. Currently, animal experiments are suspended in our research institute and the possibility to entering the lab is limited due to the pandemic restrictions.
Anyway, we would like to emphasize the following considerations.
- sEVs are secreted into blood by the various cells from the different mouse organs with a likely different secretion rate, which is regulated by a secretion/clearance balance and appear to be measurable only using a specific highly sensitive sEV labelling technology (e.g. Matsumoto et al., 2019 doi.org/10.1080/20013078.2019.1696517).
- sEVs released in the mice blood by the human tumor cells injected in the mouse brain are indistinguishable as size and morphology from the sEVs of mouse origin.
- We think that the estimation of the 10% of human sEVs in the mouse plasma we obtained by Immunocapturing the sEVs of human origin and detecting them by the human-specific anti-human CD81 antibody (see Fig. S1 of the revised version) was a reliable method to identify the sEVs definitely released by the human tumor cells injected in the mice.
2. Figure 3A : please show wider “overview” images for the analysis of sEVs by TEM, only one sEV is shown per picture, which is not conclusive. The EVs are very small (average 52 nm). Pleas add a ref for this statement (line 128, 129).
It was not suggested to replace the figure but to show a wider overview TEM picture in addition.
AR2: Following the reviewer’s suggestion, we included also a wider overview TEM image in the new version of Fig. 3A.
3. The differences between the enrichment scores obtained in Table 1 and Table 2 are not discussed. Both Tables could also move to the Supplements as they are not very informative. Additionally, the whole analysis of miRNA target genes and their potential involvement in specific pathways or functions should be extended to include Targetscan and should be discussed more critically. The target genes are predicted and as we now know, there are many false positive genes in these lists. This could be critically acknowledged in the discussion.
AR1: The enrichment scores reported in the supplementary Table S3 and Table S4 of the revised version are calculated for each annotation cluster by DAVID tool, using the Functional Annotation Clustering analysis option. They are the overall enrichment scores for the annotation clusters based on the EASE score (a modified Fisher Exact P-value) of each term member. The higher, the more enriched.
The enrichment scores shown in Table S4 are higher than those in Table S3 because the target genes used as input list for Table S4 enrichment analysis are much more than target genes used for Table S3 enrichment analysis. Consequently, the p-values of the enrichment of each annotation cluster member are more significant and the global score of the annotation cluster is higher.
To better describe this functional annotation clustering analysis, we added a new paragraph in Materials and Methods section, titled “4.13. Identification of miRNA target genes and functional enrichment analysis” at lines 600 onwards, and added a new reference [N.101 and 102] in Bibliography.
Line 600? Shows 4.8. Light and electron microscopy ??.
AR2: The reason why the reviewer did not find the paragraph at the indicated lines could be that he was reading the revised version of the paper in PDF format. The line numbers we indicated are referred to the Word document of the revised version. Now is line 626 in the Word document of the revised version Round2.
As suggested, we moved the two tables to Supplementary Table S3 and S4, respectively.
As properly reminded by the reviewer, we well know the issues about miRNA target prediction. For this reason, we did not use traditional target prediction tools, such as TargetScan, MiRanda, PITA, but we chose to use DIANA TOOLS TarBase, which is a manually curated database collecting only experimentally validated targets derived from specific, as well as high-throughput experiments, such as microarrays, proteomics, HITS-CLIP and PAR-CLIP (Vergoulis, Nucl Acids Res 2012). To better describe this database, we added a new paragraph in Materials and Methods section, titled “4.13. Identification of miRNA target genes and functional enrichment analysis” at lines 600 onwards, and added a new reference [N.101 and 102] in Bibliography.
4. In this context, Lines 328-350 : the genes found in the enrichment analysis (KECG pathways) are discussed but the specific miRNAs targeting these genes are not specified at all. Please mention which miRNAs targets the mentioned genes.
AR1: Thanks to the reviewer for this comment. We noticed that most of the target genes mentioned in the discussion were targeted by the miR-224-5p. This miRNA was initially classified as tumor suppressor miRNA, but studying more accurately its targets it is emerging that miR-224-5p could also display an oncogenic function. It could be one of those miRNAs with a dual function of tumor suppressor and oncogene. On the other hand, there are already some indication in literature about its possible role as oncomiR (references N. 88-90). Our data also suggest this dual function. Indeed, miR-224-5p targets both genes with tumor suppressor and genes with oncogenic function. Very interestingly we found this gene more expressed in the EVs from the metastatic patients and absent in the healthy samples from the in silico analysis.
All these consideration and new text along with new bibliography have been added in the new version of the manuscript, lines 386-392.
Something is missing in the revised manuscript. Lines 359-404 are not there.
AR2: this evidently happened during the conversion from Word document to PDF format made by the journal. We did not make this conversion. However, if comparing the two documents, there is only a mismatch in line numbers but the text has not been lost. We suggest to the reviewer to read the Word document we uploaded, where the lines indicated are correctly numbered.
5. A validation by individual qPCRs should be performed in order to confirm the expression level values obtained for the most interesting miRNAs by RNA sequencing. This should be feasible as only few miRNAs were highlighted.
AR1: We do not agree with the reviewer that our RNA-seq data must necessarily be validated by RT-PCR. This is based on the following-reasons. (i) The NGS data regarding levels of gene expression are quite robust and, unlike NGS-identified gene mutations, it is now widely recognized that they must not be necessarily validated with an orthogonal method. (ii) In the case of circulating and EV-carried miRNAs, widely recognized “housekeeping“ miRNAs are not available for use as RT-PCR control reference. (iii) Many of the data presented in this manuscript are based on the differences between the human and the mouse sequence of orthologs miRNAs, and we know that the RT-PCR technology does not always guarantee this possibility.
6. The authors did not explain the absence of miR-23a-3p, the most abundant miRNA in the sEVs cultivated in vitro but absent of the list of miRNAs obtained from the in vivo A validation of the expression of miR-23a-3p by qPCR should also be done for the sEVs obtained in vitro and in vivo with the orthotopic melanoma xenograft model.
AR1: It should be noted that miR-23a-3p, most abundant in the sEVs released by cultured cells (Fig. S3 of the revised version), is among the most abundant conserved miRNAs both in the total plasma and in the EVs (Fig. 6). But not present amongst human miRNAs in Fig.5?
AR2: The miRNAs shown in Figure 5 are those classified as human-specific according to the bioinformatics pipeline described in the paper. Since miR-23a-3p is classified as conserved, it appears among the conserved miRNAs shown in Figure 6 and not among the human-specific miRNAs shown in Figure 5.
The class of conserved miRNAs already pre-exist at relevant levels in the mouse organism, not allowing an accurate analysis of their possible changes. Nonetheless, we analysed only those conserved miRNAs that were up-regulated during tumor growth (considering the increase over the control mice), assuming that such miRNAs were likely released by tumor cells and this could be the case of miR-23a-3p. However, it is not surprising that cells in culture behave differently from the same cells injected as xenograft in mice. In vivo, miRNAs regulating gene expression and influencing malignant cell behaviour are in turn influenced by external stimuli from the tumor microenvironment, such as acidic extracellular conditions, presence of hormones, cytokines, external physical stimuli or hypoxia. In the revised version of the manuscript to clarify this point, we added a paragraph, along with relative bibliography, in the discussion, line 355-360 and references N. 72-76.
The lines seem to be shifted.
AR2: Please see answers to point 3 and 4.
In addition, miR23a-3p has been also validated in the case series from the public available dataset (Fig. 7A and B). Therefore, we think that qPCR validation is not necessary for the same reasons already explained in the answer to point five. Moreover, the authors think that understanding why some miRNAs are differently expressed between in vitro and in vivo sEVs would require further experiments to be considered in a future work.
The new data from healthy controls (Fig 7B) show very similar expression patterns for many of the depicted miRNAs compared to the stage IV samples, which defers the point of potential as biomarkers. Can this be explained or the miRNAs that are slightly different between A and B could be highlighted. As a suggestion: and average expression could be shown with error bars illustrating the 8 samples and then A and B could be combined in 1 graph with the 2 bars (stage IV and healthy) next to each other.
AR2: We are very grateful to the reviewer for his comment. His suggestion helped us to better define the miRNAs packaged in the sEVs and secreted by the melanoma cells injected in mouse brain. We better focused the miRNAs that could be related to melanoma progression. Following the reviewer’s suggestion, we combined Fig. 7A and B in a single barplot (now Fig. 7A) showing the mean miRNA expression in each sample group (metastatic cases in red and healthy controls in blue) with error bars of standard error. It must be reminded that log2 transformation of intensity values, here needed to plot together miRNAs with very different expression levels, tends to shrink differences among samples and to make expression levels look more similar than they really are. In fact, if looking at the boxplot in the new panel B of the same Fig.7, expression differences among sample groups appear more evident. We proposed as potential candidate biomarkers of melanoma progression only these three miRNAs in Fig.7B (miR-224-5p, miR-130a-3p and miR-21-5p) as they showed a trend of up-regulation from healthy controls to primary up to metastatic melanoma cases.
The way the different clusters were obtained in Table 1 and table 2 is not explained.
AR1: For both Table S3 and Table S4 of the revised version, the functional enrichment analysis was performed by DAVID tool using the Functional Annotation Clustering analysis option. This detail has been added in a new paragraph in Materials and Methods section, titled “4.13. Identification of miRNA target genes and functional enrichment analysis” at lines 600 onwards.
(?)
AR2: Please see answer to point 3 and 4.
Table S3 includes genes targeted by human-specific miRNAs found expressed in total plasma and sEV (Fig5B-left and right panel, respectively), Table S4 includes genes targeted by conserved miRNAs found expressed in total plasma and sEV (Fig6B-left and right panel, respectively).
8. The analysis of EV-associated miRNAs in human metastatic melanoma cases, the number of samples is very low and thus cannot provide significant results. It would have been interesting to compare your data with other public available datasets of miRNA expression detected in sEVS from other melanoma cell lines (metastatic or not). In addition, the choice of miRNAs in this analysis is not clear and is confusing for the reader. MiR-224-5p appears in Figure 5 while miR-130a-3p, miR-21-5p, miR-23a-3p, miR-29b-3p are present in the sEVs values of Figure 6B and miR-29a-3p is shown in the plasma values of Figure 6B.
AR1: We agree with the reviewer but, at the best of our knowledge, no studies are reported showing miRNA expression in sEV from orthotopic model of metastatic melanoma in the brain, so the comparison of our results with melanoma cells in culture should be misleading in consideration of the cell heterogeneity. At the same time, we would like to outline that, by our model, it is possible to identify candidate circulating miRNA packaged in EVs, which are specifically released by tumor cells and that this model could be very useful to progress in this still largely unexplored field of investigation. These considerations are discussed in the revised manuscript. Moreover, we chose to validate all the miRNAs expressed in sEVs (that is, over 2% of threshold expression level), both human-specific (Fig.5B-right panel, 9 miRNAs) and conserved (Fig.6B-right panel, 16 miRNAs). The selection criteria for the choice of miRNAs of interest is described at line 251 onwords. Regarding miR-29a-3p, it is expressed in total plasma (Fig.6B-left panel), but also in sEVs of our experimental model, as reported in Fig.6B-right panel, and therefore we included it in the validation subset.
9. Functional assays to explore the potential roles of miRNAs which are enriched in sEVs derived from melanoma cells or which are up-regulated during tumour growth would give the findings more weight.
AR1: We agree with the reviewer that experiments on the functionality of the different miRNAs could be of great interest. However, we think that these studies are not the focus of this paper that rather aims to assess how the use of an experimental model of xenograft in vivo can be applied to study sEV-associated miRNAs released by a metastasis at the brain.
10. The roles of the most abundant human specific miRNAs in sEVs ; miR-6131, miR-6853-5p, miR-1268a, miR-4258, are not discussed and should be tested experimentally. Their target gene are not mentioned.
AR1: We agree with the reviewer that the functional role of the most abundant human specific miRNAs in the sEVs should be of interest, but as we already mentioned in the discussion, the understanding of novel functions associated with sEV-enriched miRNAs is beyond the scope of this paper and will be further investigated in a future work. Hence, we did not mention the targets of the two most abundant sEV miRNAs as they are not validated yet, they are only predicted targets.
But they should nevertheless be discussed as they are the most abundant miRNAs.
Tables S3 and S4 should indicate the groups of miRNAs that were used to generate the pathway annotations.
AR2: As requested by the reviewer, in the new revised version of the manuscript we added text to discuss the most abundant sEV human-specific miRNAs (lines 409-417 in the Word document of the revised R2 version) and in Table S3 and S4 we added the miRNAs used to generate the pathway annotations.
11. The discussion could be shortened and more focused on the purpose of the study, i.e. identification of miRNA biomarkers. The herein found miRNAs should be discussed in more detail in the context of previously published data.
AR1: The overall discussion has been shortened even though new paragraphs have been added to address the referees’ requests.
12. In this context, some references should be included while others seem out of place or not suitable for the statement. References that could be mentioned: Gowda R. et al., Cancer Treat Rev, 2020; Carpi, S. et al., Exp Rev Mol Diagn. 2020; Cesi, G. et al., Cell Comm Signal 2016; Margue et al., Oncotarget 2015 and others that are covering the miRNA/exosome/biomarker topic.
AR1: We added in the discussion the bibliography suggested by the reviewer. Lines 279-281, references n: 26-29..
13. Lines 292-294 : The authors should discuss that the presence of ZONAB and GFAP in sEVs was already mentioned in two previous publications, not at the membrane and neither by the same experimental procedure (doi : 10.1074/mcp.M112.022806, doi : 10.1016/j.jprot.2012.12.029, doi : 10.18632/oncotarget.3801, doi: 10.1074/mcp.M900381-MCP200).
AR1: As suggested by the reviewer, we discuss now some of the above mentioned papers (lines 323-324) and added them to the reference list (references N. 54, 55).
14. Line 158 onwards: the control sample had no tumor cells injected, so should not have human sequences but the heatmap only shows human miRNAs. This should be better explained or corrected. Overall, a Table would useful to show the numbers of sequence reads for the 3 classes: mouse miRNAs, human miRNAs, both.
AR1: The heatmap reports the actual data obtained from the sequence analysis, without any additional filtering other than the sequence quality score, and notwithstanding, as described in M&M, the reads were directly mapped on miRBase with a mismatch value of 0, the very low signals shown in the control samples by some human specific miRNAs should be accounted to sequence errors.
Following the reviewer’s suggestion, normalized read counts (CPM) obtained for the miRNAs of the three datasets found in mice total plasma as well as enriched in sEVs have been provided in Supplementary Table S1 and S2, respectively (lines 165-167).
Please reformat the supplementary tables to make them easier to read. Table S1 and S1-1 and S2: please combine the values of the different time points in 1 spreadsheet and separate tables for the mouse-, human- specific and conserved miRNAs.
AR2: Following the reviewer’s suggestion, we have reformatted the supplementary Table S1 and S2 as indicated.
15. Figure 4: labels are too small.
AR1: We apologize with the reviewer for this inconvenience probably due to the copy of the figure in the word document of the manuscript. For sharper image, we upload the TIFF document of Fig. 4
16. Line 266: Why “On the other hand”?
AR1: We agree with the reviewer’s comment since it seems more proper to change "On the other hand" to "In addition". Change have been made in the new version line 291.
17. A good experimental control would be the sequencing of tissue-miRNAs in the brain metastasis of mice (after sacrificing them) to have a comparison between tissue-based miRNAs and secreted miRNAs in plasma.
AR1: As suggested by the reviewer, we added in the manuscript new data obtained by miRNA profiling by qRT-PCR, using TaqMan Human Array MicroRNA A+B Cards of M14 melanoma tumor xenografts from mice brain at day 23 after cell injection. Being a technology different from next-generation sequencing, a direct comparison between tissue-based miRNAs and circulating miRNAs found in our experimental model is not feasible. Nonetheless, we can evaluate in these brain tumor xenografts the expression of those total plasma and sEV-enriched miRNAs passing the expression threshold of 2% (those shown in Fig.5B and 6B). These new findings are now described in the manuscript at lines 241-246, and in the new Figure S4.
As there are no error bars, can the authors indicate whether replicates have been performed and how many samples have been tested.
AR2: The experiment has been performed by pooling xenografts from 4 mice (so considering the variability among the different mice) each in duplicate. As requested, standard error bars have been added in the graphs shown in Fig. S4. In the materials and methods section, information about the experiment performed was already present in the revised first round version of the manuscript (paragraph 4.15, lines 645-654 of the revised version Round 2- Word document) as well as in the Figure S4 legend.